# Partially overlapping spatial environments trigger reinstatement in hippocampus and schema representations in prefrontal cortex

Li Zheng [1,2], Zhiyao Gao [3], Andrew S. McAvan[1,2], Eve A. Isham[1,2] & Arne D. Ekstrom [1,2✉]

When we remember a city that we have visited, we retrieve places related to finding our goal but also non-target locations within this environment. Yet, understanding how the human brain implements the neural computations underlying holistic retrieval remains unsolved, particularly for shared aspects of environments. Here, human participants learned and retrieved details from three partially overlapping environments while undergoing high-resolution functional magnetic resonance imaging (fMRI). Our findings show reinstatement of stores even when they are not related to a specific trial probe, providing evidence for holistic environmental retrieval. For stores shared between cities, we find evidence for pattern separation (representational orthogonalization) in hippocampal subfield CA2/3/DG and repulsion in CA1 (differentiation beyond orthogonalization). Additionally, our findings demonstrate that medial prefrontal cortex (mPFC) stores representations of the common spatial structure, termed schema, across environments. Together, our findings suggest how unique and common elements of multiple spatial environments are accessed computationally and neurally.

[1] Department of Psychology, University of Arizona, 1503 E. University Blvd., Tucson, AZ 85721, USA. [2] Evelyn McKnight Brain Institute, University of Arizona, 1503 E. University Blvd., Tucson, AZ 85721, USA. [3] Department of Psychology, University of York, Heslington, York YO10 5DD, UK.
✉email: adekstrom@email.arizona.edu

When asked to think about the dairy section in a familiar supermarket, other spatially proximate surrounding sections such as the frozen foods aisle often come to one's mind incidentally. This reinstatement of neighboring elements within the same spatial environment is thought to be a fundamental property of spatial representations termed "holistic retrieval"[1], underlying our ability to forage for food, plan future behavior, and even avoid threats in some situations. Human neuroimaging studies of episodic memory retrieval support the idea of reinstatement[2–4] (i.e., the activity patterns of a previous experience reoccur when triggered by a partial cue), incidental retrieval[5] (i.e., memory reinstatement not specific to the target item but also involving some of the neighboring elements bound to that target), and transitive inference[6–8] (i.e., studying A-B and B-C, then inferring A-C). While these likely have some commonalities with holistic retrieval, it is unclear how the recovery of specific events relates more broadly to retrieval of spatial environments, which we often experience at multiple time points and in different manners at different times. In addition, an unresolved issue about holistic retrieval in spatial environments, in particular, is how the brain processes shared information between two (or more) different environments that would lead to conflicts during holistic retrieval. In other words, if one shops at two supermarkets in the same city that contain the same sections (i.e., dairy and frozen sections), how does one avoid interference when reinstatement of two different supermarkets is triggered by the same sections that they share in common?

The issue of holistic retrieval vs. managing interference from shared components will typically be less of an issue for episodic memories and specific routes because they involve unique moments in time[9,10]. This potentially allows for disambiguation of events[11] but is potentially catastrophic to more generalized spatial memories. If both supermarkets have dairy sections in the same locations yet frozen food locations in different places, standing in front of the dairy food section in supermarket A may result in retrieving the frozen food section for supermarket B. This confusing information would result in the incorrect navigational representation for finding the frozen food section and other locations in supermarket A. One candidate mechanism within the hippocampus that could play a role in balancing interference for similar inputs is pattern separation, a computational process involving orthogonalizing similar inputs[12–14], with pattern separation suggested to occur in hippocampal subfields CA3 and/or DG[13,15–19]. Single-neuron studies demonstrate that place cell "remapping," in which place cell firing patterns show a near-zero correlation between two (in some cases similar) environments, could relate to such pattern separation mechanisms[20]. Recently, by utilizing multivariate pattern analysis (MVPA) on a population of neurons or local field potential (LFP) in hippocampus, researchers found that both the pattern of neural firing rates[21] or LFP signals[22] could support such a putative pattern separation mechanisms. Here, we leverage MVPA and high-resolution magnetic resonance imaging (fMRI) to test novel hypotheses derived from past single neuron[21], LFP[22], and fMRI studies[15,23] regarding how pattern separation might relate to the issue of environment-specific codes through mechanisms by which inputs that share some similarities—like supermarkets—can be stored separately.

Yet while pattern separation may allow for orthogonalization based on maximizing differences (i.e., one supermarket is different than another), it is not clear how holistic retrieval for a single environment can be balanced with pattern separation for different environments, particularly when they share common elements. Specifically, pattern separation may be insufficient for representing otherwise identical elements between two different spatial representations (i.e., the dairy sections in two different supermarkets). One potential solution comes from recent fMRI studies showing that hippocampal activity patterns can form "reversed" (past the point of orthogonalization) representations between shared elements, suggesting a repulsion mechanism of eliminating overlapping memory representations[24–28]. According to this mechanism, the similarity of overlapping sections between Supermarket A and B would be repulsed to a greater extent, exhibiting reverse similarity between overlapping sections compared to the similarity of two different sections (i.e., the similarity between two food sections should be lower than the similarity between the food sections and the parking lot). We leveraged these past findings related to repulsion to test novel hypotheses based on the assumption that "reversed" pattern similarity past the point orthogonalization — could be a potential mechanism for maintaining overlapping representations. Although one study suggested that hippocampal representations of overlapping routes became more dissimilar than non-overlapping routes during later learning stages[24], it remains unclear whether "repulsion" also occurs during spatial memory retrieval of competing spatial environments. In addition, given the heterogeneous nature of the hippocampus[14,29,30], it is important to resolve the potentially different roles played by the hippocampal subfields.

One possible candidate brain area for modulating hippocampal-mediated processing during spatial retrieval of competing information is the lateral prefrontal cortex (PFC), which may contribute to accurate retrieval via control processes[31–33]. On one hand, the lateral PFC could accentuate retrieval of some responses and memories[34] while on the other hand, the lateral PFC could play a role in suppression of unwelcome and competing information during memory retrieval[35–38]. Schematic "generalized" representations of the shared elements of multiple environments, potentially represented in medial PFC (mPFC), could provide one means by which locations that share overlap vs. those that are unique can be "tagged." It is not clear, however, whether the mPFC or the hippocampus processes such schema. Cognitive map theory emphasizes the formation of an integrated map to infer spatial relationships among elements, and, in this way, a cognitive map shares many similarities with spatial schemas[39], with some studies suggesting that the hippocampus may serve this function[40–42]. On the other hand, numerous empirical and theoretical papers support a role for the mPFC in schematic representations[43–47]. Past studies have also suggested that hippocampus-mPFC interactions are important to the integration of multidimensional cognitive maps, including spatial[48–50], temporal[51–56], social[57], and conceptual associations[8,58,59]. Whether mPFC or hippocampus plays the primary role in schema representation remains unclear, particularly regarding the different roles the two regions might play for overlapping components of spatial environments.

In the current study, we test the roles of the hippocampus and PFC in the retrieval of spatial environments that share some common elements by using a paradigm adapted from a past study[15] to create overlapping environments. Here, we employ a high-resolution fMRI approach allowing us to investigate not only hippocampal subfields but also the role of PFC during spatial memory retrieval. Participants learned three overlapping cities made up of shared stores between cities and unique stores specific to a city by playing a virtual reality navigation game (Fig. 1a, d). Participants also learned the temporal intervals between different routes, allowing us to look at both spatial and more episodic-like representations. Then, all participants were asked to retrieve details about each virtual environment they had visited by judging the spatial distance or temporal intervals between stores while their brain activity was monitored using fMRI (Fig. 1e). Critically, using a multivariate pattern similarity analysis (MPS)[60,61], we

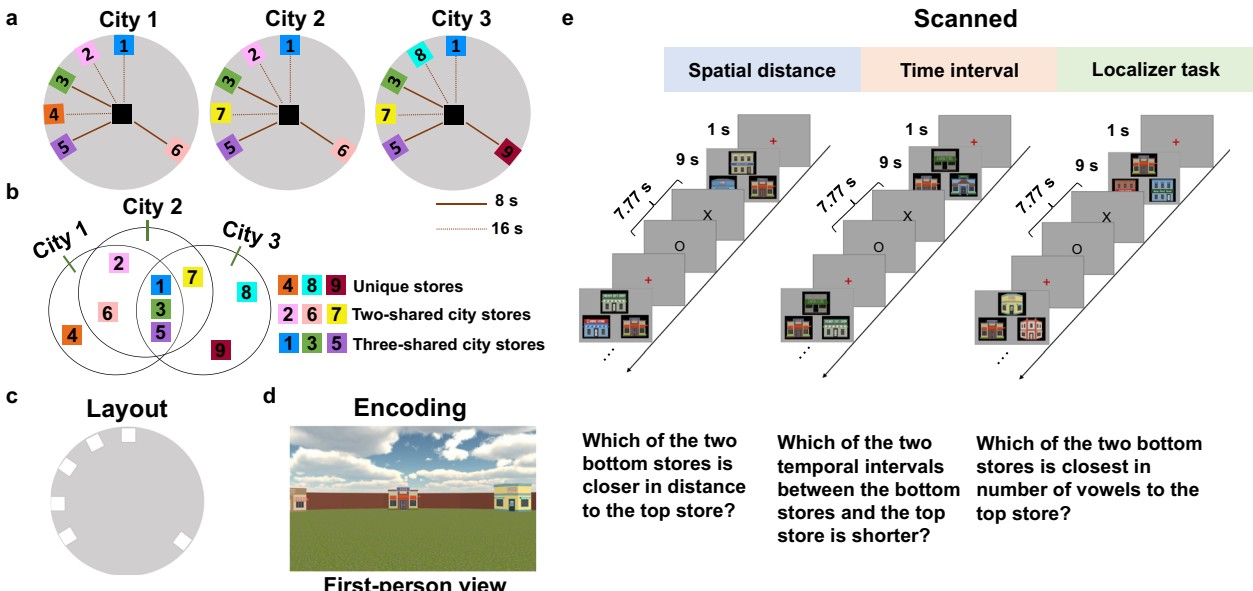

**Fig. 1 Schematic of the experimental design. a** Layouts for each environment are depicted from a survey view. Each colored box represents a target store (except the center one). Cities 1 and 2 are identical aside from the "Store 4" in City 1, which was changed to "Store 7" in City 2. Cities 2 and 3 were the same except "Store 2" and "Store 6" in City 2 were changed to "Store 8" and "Store 9" in City 3, respectively. The lines from the center store to each peripheral store represent the temporal durations: 8 s (red solid line); 16 s (red dashed line). **b** Venn diagram depicting the shared stores between the three overlapping environments (note: this does not show physical overlap but rather the stores that are the same between environments). "Unique city stores" refers to the stores that only belonged to one city (i.e., "Store 4", "Store 8", and "Store 9"); "Two shared city stores" refers to stores that belong to two cities at the same time (i.e., "Store 2", "Store 6" and "Store 7"); "Three shared city stores" refers to stores that belong to three cities at the same time (i.e., "Store 1", "Store 3", and "Store 5"). **c** The three environments shared the same physical layout of stores although the identities of stores differed. **d** An example of an encoding trial from a first-person perspective. Participants encoded the locations of stores and the duration from the center store to peripheral stores. **e** Retrieval consisted of 6 runs of city-specific spatial distance judgments and 6 runs of city-specific temporal durations judgments. After the retrieval task, participants were asked to complete a localizer task involving a vowel counting task.

construct a multivariate pattern template for each store from the localizer task, and test the holistic retrieval hypotheses by examining whether or not other non-target stores in the environment are reinstated when retrieving the target stores in a trial-specific question. This allows us to further test whether unique stores are more central to holistic retrieval rather than shared stores. We further examine whether the hippocampal reinstatement could be mediated by lateral PFC activity through functional interactions. Finally, we test whether the shared information across similar environments might be differentiated by pattern separation and/or repulsion in hippocampal subfields, while at the same time, whether they might be abstracted to form a spatial schema in mPFC or in hippocampal subfields.

## Results

**Holistic representations in hippocampus.** We sought to test whether distributed neural codes within the hippocampus might reveal evidence for multivariate codes related to holistic retrieval. In other words, when participants retrieve a neural representation for an environment, do such codes also contain information relevant to other stores in the environment as a whole in addition to those involved in the particular retrieval question? If the spatial retrieval is specific to a trial (i.e., non-holistic), we predict the correlation between that trial and "incidental" unpresented stores specific to that city should be comparable to the correlation between that trial and unpresented stores belonging to other cities (Fig. 2a, left panel). In contrast, if spatial retrieval is holistic, we predict a higher correlation between neural patterns retrieved for a specific trial and unpresented stores for that same city (i.e., unique/shared store within-city PS) compared to those specific to other cities (i.e., unique store between-city PS; Fig. 2a, middle

panel). A corollary to this is whether the "incidental" stores retrieved (i.e., those not in the retrieval question) are unique to that environment (differentiated) and not the ones shared across environments (non-differentiated). We lay out the possibilities for shared stores (i.e., holistic and non-differentiated vs. holistic and differentiated) in Fig. 2a.

We applied MPS[60,61] to test this assumption. First, for each of the six studied stores in the localizer task (i.e., Store 2, Store 4, Store 6, Store 7, Store 8, and Store 9; see "Materials" and Fig. 1b), we constructed a multivariate pattern template for that specific store based on the elicited voxel patterns within a specific region of interest (ROI). Based on which city or cities each store appeared in, all six stores could be divided into unique city stores (only belonged to one city) and stores shared between cities (belonged to two cities at the same time, Fig. 1b). The multivariate template of each store from the localizer task provided an independent measure of the contents retrieved during distance/duration questions (Fig. 1e). We conducted a three-way repeated-measure ANOVA, with the factors of 6 ROIs of MTL, 8 t-stat thresholds (tSNR) levels, and three conditions (unique store within-city PS/shared store within-city PS/unique store between-city PS) as within-subject variables. The results revealed a significant ROI-by-condition interaction ($F(5.527, 143.692) = 4.960$, $p < 0.001$, $\eta^2_p = 0.160$, Greenhouse-Geisser corrected, Supplementary Fig. 1a). Because there was no significant interaction of tSNR by ROI by condition ($F(7.647, 198.825) = 1.579$, $p = 0.136$, $\eta^2_p = 0.057$, Greenhouse-Geisser corrected), we averaged pattern similarity for each ROI with different tSNR levels for the following analyses. Simple effects revealed that the within-city PS for unique stores was significantly higher than between-city PS for unique stores both

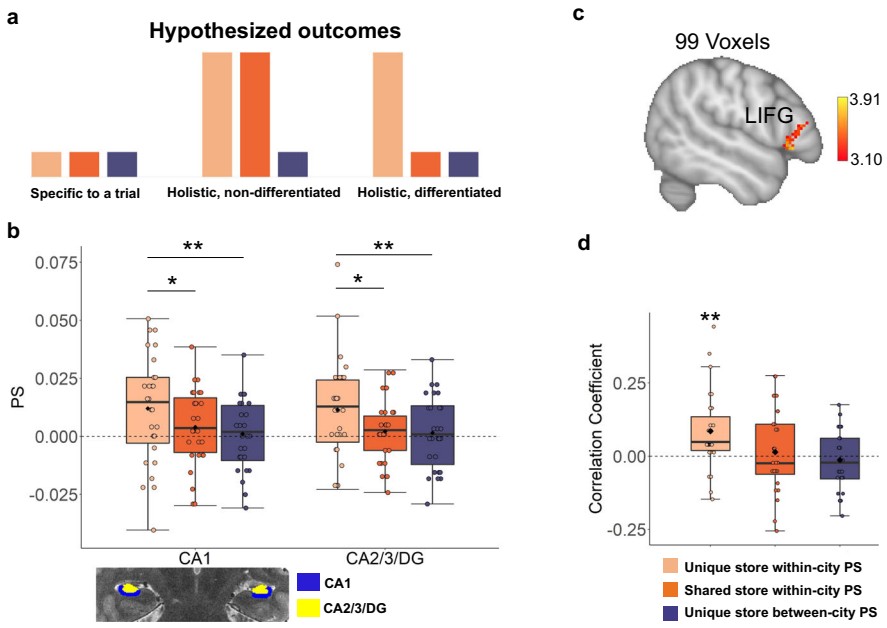

**Fig. 2 Multivariate pattern similarity analysis (MPS) of holistic retrieval in the hippocampus. a** Hypothesized outcomes of neural representations. First, if spatial retrieval is specific to a trial (i.e., non-holistic), we predict the correlation between that trial and "incidental" unpresented stores specific to that city (i.e., unique/shared store within-city PS) should be comparable to the correlation between that trial and unpresented stores belonging to other cities (i.e., unique store between-city PS, left panel). Second, if spatial retrieval is holistic, we predict a higher correlation between that trial and unpresented stores specific to a city compared to those unique to other cities (middle panel). Third, if holistic retrieval is differentiated, we predict a higher correlation between that trial and the stores unique to a city compared to those shared across cities and those unique to other cities (right panel). **b** Top: holistic and differentiated representations in hippocampus during retrieval (averaged across 8 tSNRs). Consistent with the third hypothesized outcome, the neural representations in CA1 and CA2/3/DG were holistic (CA1: $p = 0.005$, CA2/3/DG: $p = 0.005$, two-tailed paired-sample t-test with FDR correction) and differentiated from the other city representations (CA1: $p = 0.021$, CA2/3/DG: $p = 0.022$, two-tailed paired-sample $t$-test with FDR correction). Bottom: regions of interest included CA1 (blue) and CA2/3/dentate gyrus (CA3/DG, yellow) in the hippocampus from a sample participant. **c** The ventrolateral PFC (LIFG, MNI, −52, 22, −2, $Z = 3.91$) activity showed a significantly positive correlation with the PS in CA2/3/DG, after controlling for the activation levels in CA2/3/DG (a random-effects model was used for group analyses across the whole brain using a cluster-forming threshold of $Z > 3.10$, with $p < 0.05$ (corrected for family-wise error rate, using random field theory). **d** The partial correlation between LIFG activity and CA2/3/DG in each of the three conditions (unique store within-city PS/shared store within-city PS/unique store between-city PS). There was a positive correlation between the activation level of LIFG and unique-store within-city PS ($p = 0.009$, two-tailed one-sample $t$-test compared to zero which survived FDR correction). Notes: Boxplots are centered on the median, boxes extend to first and third quartiles, whiskers extend to 1.5 times the interquartile range or minima/maxima in the absence of outliers. Each individual dot represents data from an individual subject. Each black solid diamond represents the mean of the group. All data reflect $n = 27$ independent participants. LIFG left inferior frontal gyrus, PS pattern similarity. *$p < 0.05$, **$p < 0.01$. Source data are provided in the Source Data file.

in CA1 (t(26) = 3.554, $p = 0.005$, CI = [0.011 ± 0.006], Cohen's $d = 0.684$) and CA2/3/DG (t(26) = 3.475, $p = 0.005$, CI = [0.010 ± 0.006], Cohen's $d = 0.669$, FDR correction was performed for six comparisons [ROIs]) whereas the within-city PS for shared stores was not significantly different from between-city PS for unique stores (CA1: t(26) = 1.833, $p = 0.078$, CI = [0.003 ± 0.003], Cohen's $d = 0.353$; CA2/3/DG: t(26) = 0.436, $p = 0.666$, CI = [0.001 ± 0.004], Cohen's $d = 0.084$; Fig. 2b). These findings suggested that spatial retrieval was holistic and specific to the unique information of an environment (and not the shared information).

Importantly, the differences between unique store within-city PS and unique store between-city PS could not be accounted by a negative between-city PS for unique stores (see Supplementary Table 3) or the semantic association between stores (see Supplementary Note 3). Furthermore, within-city PS was higher for unique stores than that for shared stores in CA1 (t(26) = 3.133, $p = 0.021$, CI = [0.008 ± 0.005], Cohen's $d = 0.603$) and CA2/3/DG (t(26) = 2.728, $p = 0.022$, CI = [0.009 ± 0.007], Cohen's $d = 0.525$, FDR correction was performed for 6 comparisons [ROIs]), supporting the idea that such holistic representations were also differentiated from other city representations. Notably, the differences between any of the three conditions (unique store within-city PS/shared store within-city

PS/unique store between-city PS) could not be accounted by differences in univariate activation levels to stores (Supplementary Note 3) or unequal numbers of trial pairs (Supplementary Fig. 2c, see also Supplementary Note 4). Finally, we also examined whether holistic representations are also present outside the MTL by using both ROI-based and searchlight-based MPS analyses (Supplementary Note 3). No ROIs or clusters showed holistic representation-related effects outside the MTL. Therefore, all further analyses were focused on hippocampal subfields. In sum, these results show that spatial memory representations for unique city information in both CA1 and CA2/3/DG are holistic and environment specific, with shared store information treated differently than stores unique to an environment.

**PFC contributes to accentuating representations of landmarks unique to an environment.** An important question regards how interference from stores from shared cities is minimized while those unique to a city are successfully retrieved. Understanding this mechanism is also important to understanding how interference between shared city features can be minimized during retrieval. One candidate region is the lateral PFC, a key region thought to directly modulate hippocampal representations[62,63]. For example, lateral PFC could contribute, via functional

interactions, to the retrieval of context-appropriate memories by suppressing competing, context-interfering memories[35,64]. If so, we predict that prefrontal activity should be different for the retrieval of overlapping environments by enhancing the city-unique representation and suppressing shared city representations and those unique to other cities in hippocampus.

We conducted a partial correlation analysis using the unique store within-city PS, shared store within-city PS, and unique store between-city PS within hippocampal subfields. We correlated these measures with the levels of activation across the brain while controlling for the activation level of hippocampus subfields (see "Methods"). This allowed us to detect, in an unbiased manner, which voxels outside of the hippocampus might be modulating hippocampal environment-specific signals. We chose CA2/3/DG and CA1 as regions of interest in this partial correlation analysis because these two subfields were associated with successful memory retrieval in our previous analyses. When using CA2/3/DG as the region to be modulated, a direct comparison between unique store within-city PS vs. unique store between-city PS revealed stronger modulation of ventrolateral PFC (left inferior frontal gyrus [LIFG], MNI: −52, 22, −2, $Z = 3.91$), which was greater for within than between-city PS (Fig. 2c). By extracting the averaged partial correlation coefficient (and employing the Fisher's z-transform) of three conditions (unique store within-city, shared store within-city, and unique store between-city) from the LIFG cluster, we observed a positive correlation between the activation level of LIFG and unique-store within-city PS ($t(26) = 3.30$, $p = 0.009$, CI = [0.086 ± 0.054], Cohen's $d = 0.636$, compared to zero, two-tailed, FDR correction was performed for three comparisons, Fig. 2d). In contrast, there was no significant correlation between the activation level of LIFG and unique store between city PS ($t(26) = 0.516$, $p = 0.611$, CI = [0.014 ± 0.055], Cohen's $d = 0.099$, two-tailed, compared to zero) or shared store within-city PS ($t(26) = −0.667$, $p = 0.511$, CI = [−0.013 ± 0.040], Cohen's $d = −0.128$, two-tailed, compared to zero). However, when we used CA1 as the region to be modulated, we did not find significant clusters of activation exceeding chance anywhere within the brain. These results suggest the possibility that the PFC might modulate multivariate signals in CA2/3/DG through functional interactions resulting in prioritization of retrieval of stores unique to a city.

**Hippocampal neural codes for shared city trials involve differentiated and "repulsed" representations.** Earlier, we reported that the distributed patterns of neural activity for stores shared between multiple cities were significantly less correlated than stores that were unique to the current city being retrieved. An important next question regards the neural representation for the shared city stores: these stores were present in at least two different environments and were needed to solve specific distance questions. Yet, even when they were part of the same city, they showed a lower correlation than stores unique to that city. This raises the possibility that these shared stores were in a different representation entirely, a form of repulsion[24,25]. Alternatively, it could simply be that their neural signals were differentiated from those unique to a city but still part of that same representation.

We can test these possibilities by comparing trials that are unique to a city, trials shared between two cities, and trials shared between three cities (Fig. 3b, c, and also see "Methods"). We calculated the between-city PS within each condition: unique city trials, two shared city trials, and three shared city trials (see "Methods"). We predicted that, if the memory representation of trials shared between cities is the same, the between-city PS of shared city trials should be higher than between-city PS of unique-city trials (Fig. 3a, left panel). Alternatively, if the memory

representation of the shared city trials is orthogonal between cities (perhaps related to pattern separation), the between-city PS of shared city trials should be comparable (statistically equivalent) to the between-city PS of unique-city trials (Fig. 3a, middle panel). Finally, it could be that, as part of interference reduction, the representations for shared compared to unique city trials undergo a form of "repulsion". This would predict lower between-city PS for the shared city trials compared to trials unique to a different city (Fig. 3a, right panel).

Focusing on the 2 ROIs (i.e., CA1, CA2/3/DG) which showed significant effects in our previous analyses related to holistic retrieval, we conducted a two-way repeated measure ANOVA, with the factors of 8 tSNR levels and three conditions (between-city PS for unique city trials, between-city PS for two shared city trials, and between-city PS for three shared city trials) as within-subjects variables in each ROI separately. We found a significant main effect of condition in CA1 ($F(1.642, 42.694) = 5.481$, $p = 0.007$, $\eta^2_p = 0.174$, Greenhouse-Geisser corrected), but not in CA2/3/DG ($F(1.512, 39.305) = 0.973$, $p = 0.385$, $\eta^2_p = 0.036$). Because there was no significant interaction of tSNR by condition in the two ROIs ($Ps > 0.174$, Greenhouse-Geisser corrected, Supplementary Fig. 3), we then averaged pattern similarity for each ROI across different tSNR levels and compared the three conditions in a post hoc paired $t$-test (Fig. 3d).

We found that between-city PS for unique city trials was significantly higher than three shared city trials ($t(26) = 3.364$, $p = 0.004$, CI = [0.008 ± 0.005], Cohen's $d = 0.647$, two-tailed, FDR corrected) in CA1, suggesting a repulsion effect for three shared city trials. Between-city PS for two shared city trials, however, was not significantly lower than unique city trials ($t(26) = 0.462$, $p = 0.648$, CI = [0.001 ± 0.006], Cohen's $d = 0.089$, two-tailed) but was significantly higher than three shared city between-city PS ($t(26) = 3.066$, $p = 0.010$, CI = [0.006 ± 0.005], Cohen's $d = 0.590$, two-tailed, FDR corrected) in CA1. These findings suggest that while the three shared city trials showed repulsion compared to the unique city trials, the two shared city trials did not, an issue we return to in the "Discussion" section. CA2/3DG, in contrast, did not show a significant difference between unique city trial PS and two ($t(26) = −0.561$, $p = 0.579$, CI = [−0.002 ± 0.007], Cohen's $d = −0.108$, two-tailed) or three shared city trials PS in CA2/3/DG ($t(26) = 0.755$, $p = 0.457$, CI = [0.002 ± 0.005], Cohen's $d = 0.145$, two-tailed, Fig. 3d), suggesting pattern separation for shared city trials. The different patten between CA1 (i.e., repulsion) and CA2/3/DG (i.e., pattern separation) was confirmed by a significant ROI (i.e., CA1, CA2/3/DG) by condition (between-city PS for unique city trials, between-city PS for three shared city trials) interaction ($F(1,26) = 4.379$, $p = 0.046$, $\eta^2_p = 0.144$). Importantly, there was no main effect between the two ROIs ($F(1,26) = 0.543$, $p = 0.468$, $\eta^2_p = 0.020$), and the observed repulsion effect in CA1 (but not in CA2/3/DG) could not be accounted by higher between-city PS for unique city trials in CA1 because there was no significant difference for between-city PS for unique city trials between CA1 and CA23DG ($t(26) = 1.619 = 0.117$, CI = [0.004 ± 0.006], Cohen's $d = 0.312$, two-tailed).

As additional control analyses, we performed the same MPS analysis using the same number of trial pairs from each condition and obtained similar results (Supplementary Fig. 4c, see also Supplementary Note 5). Furthermore, considering that raw PS scores can be influenced by tSNR and univariate activation levels, we computed a relative difference score for between-city PS for three shared city trials compared to unique city trials (Fig. 3e). This result again revealed a repulsion effect in CA1 ($t(26) = −3.364$, $p = 0.004$, CI = [0.008 ± 0.005], Cohen's $d = 0.647$ compared to zero, two-tailed, FDR corrected), but not in CA2/3/DG ($t(26) = −0.755$, $p = 0.457$, CI = [0.002 ±

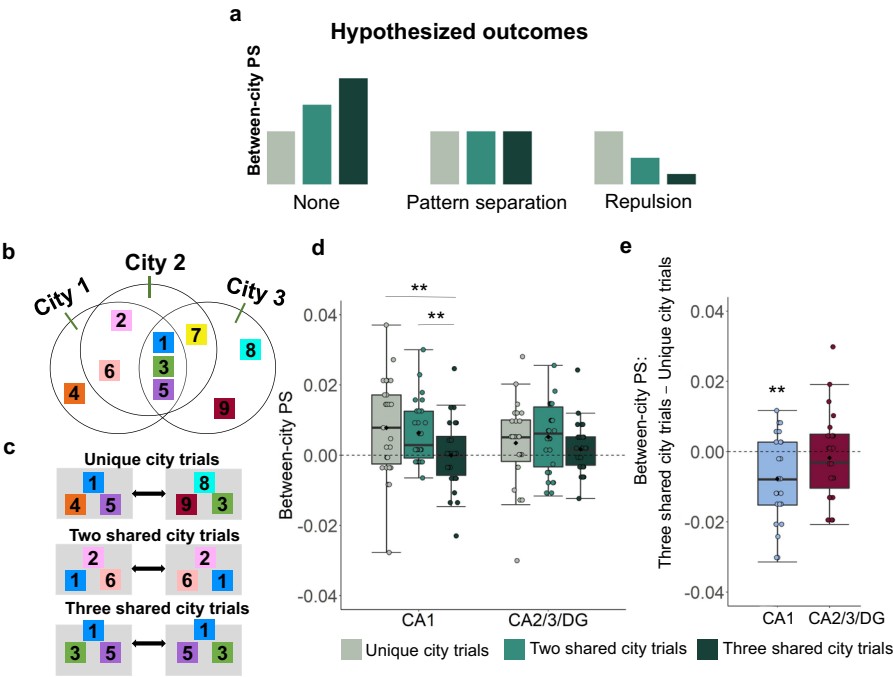

**Fig. 3 Multivariate pattern similarity analysis (MPS) of differentiated representations for shared city trials in the hippocampus. a** Hypothesized outcomes for shared city trial representations. First, if the memory representation of trials shared between cities is the same, the between-city PS of shared city trials should be higher than the between-city PS of unique city trials (left panel). Second, if the memory representation of the shared city trials is orthogonal between cities (i.e., due to pattern separation), the between-city PS of shared city trials should be comparable to the between-city PS of unique city trials (middle panel). Third, if repulsion occurs, the between-city PS for the shared city trials should be lower than trials unique to a different city (right panel). **b** Venn diagram depicting store overlap across the three environments, same as Fig.1b. **c** Examples of three kinds of shared city trials/conditions. Unique city trials (i.e., a trial that could only be attributed to one city, for example, the triads "Store 1-Store 4-Store 5" was only attributable to City 1 and the triads "Store 8-Store 9-Store 3" was only attributable to City 3, top panel), two shared city trials (i.e., trials that could be attributed to two possible cities, for example, the triads "Store 2-Store 1-Store 6" and triads "Store 2-Store 6-Store 1" could only be attributed to City 1 and City 2 but not be attributed to city 3, middle panel) and three shared city trials (i.e., trials that could be attributed to three possible cities, for example, the triads "Store 1-Store 3-Store 5" and "Store 1-Store 5-Store 3" could be attributed to City 1, City 2 and City 3, bottom panel). **d** The neural representations for three shared city trials showed a repulsion effect in CA1 (consistent with the third hypothesized outcome, $p = 0.004$, two-tailed paired-sample $t$-test with FDR correction) and a pattern separation effect in CA2/3/DG (consistent with the second hypothesized outcome, averaged across 8 tSNRs). The two city shared trials did not show lower between-city PS than unique city trials ($p = 0.648$, two-tailed paired sample $t$-test with FDR correction) but significantly higher between-city PS than three city-shared trials ($p = 0.010$, two-tailed paired-sample t-test with FDR correction). **e** The relative difference in between-city PS for three shared city compared to unique city trials ($p = 0.004$, two-tailed one-sample $t$-test with FDR correction, compared to zero). Notes: Boxplots are centered on the median, boxes extend to first and third quartiles, whiskers extend to 1.5 times the interquartile range or minima/maxima in the absence of outliers. Each individual dot represents data from an individual subjects. Each black solid diamond represents the mean of the group. All data reflect $n = 27$ independent participants. PS pattern similarity. ** $p < 0.01$. Source data are provided in the Source Data file.

0.005], Cohen's $d = 0.145$, compared to zero, two-tailed). Together, these results indicate that both CA2/3/DG and CA1 contributed to the discrimination of overlapping memories but through potentially different mechanisms. CA2/3/DG appeared to differentiate similar cities by pattern separation, which was evidenced by comparably low between-city PS for unique-city trials compared to shared-city trials. CA1, in contrast, showed higher between-city PS for unique-city trials compared to three shared-city trials, which may be a form of repulsion, suggesting that shared stores were part of a different representation.

**Schematic spatial layout representation in medial PFC.** Our previous results showed a potential modulatory role for the PFC in spatial and temporal processing when participants retrieved information about different environments. To play a modulatory role, however, the PFC would also appear to require some representation of what was shared across the different environments. Given past suggestions for a role for the medial PFC (mPFC) in schema representation[43–47], in other words, the

shared positional elements across all cities (Fig. 1c), we next tested to see whether mPFC revealed evidence for such schema.

To address this issue, we adopted a searchlight-based leave-one-city-out SVR classifier throughout the whole brain (see "Methods"). This method allowed us to determine whether spatial distances in two cities could be used to generalize spatial distances in a different city. The SVR classifier revealed two clusters whose spatial distances were able to generalize from training on two cities to a third one. This included a more superior cluster (mostly located in paracingulate gyrus, MNI: −4, 50, 18, $Z = 3.676$, Fig. 4a) and a more ventral cluster (mostly located in anterior cingulate gyrus and paracingulate gyrus, MNI: −10, 34, −4, $Z = 4.019$, Fig. 4c) in mPFC. Testing each city's classifier accuracy against chance in each cluster revealed that the classifier performed above chance on all cities in the superior cluster (City 1: t(26) = 3.031, $p = 0.005$, CI = [0.070 ± 0.045], Cohen's $d = 0.583$; City 2: t(26) = 3.887, $p < 0.001$, CI = [0.079 ± 0.045], Cohen's $d = 0.748$; City 3: t(26) = 4.346, $p < 0.001$, CI = [0.105 ± 0.045], Cohen's $d = 0.836$; two-tailed, survived by FDR correction, Fig. 4b) and the ventral cluster (City 1: t(26) = 2.881,

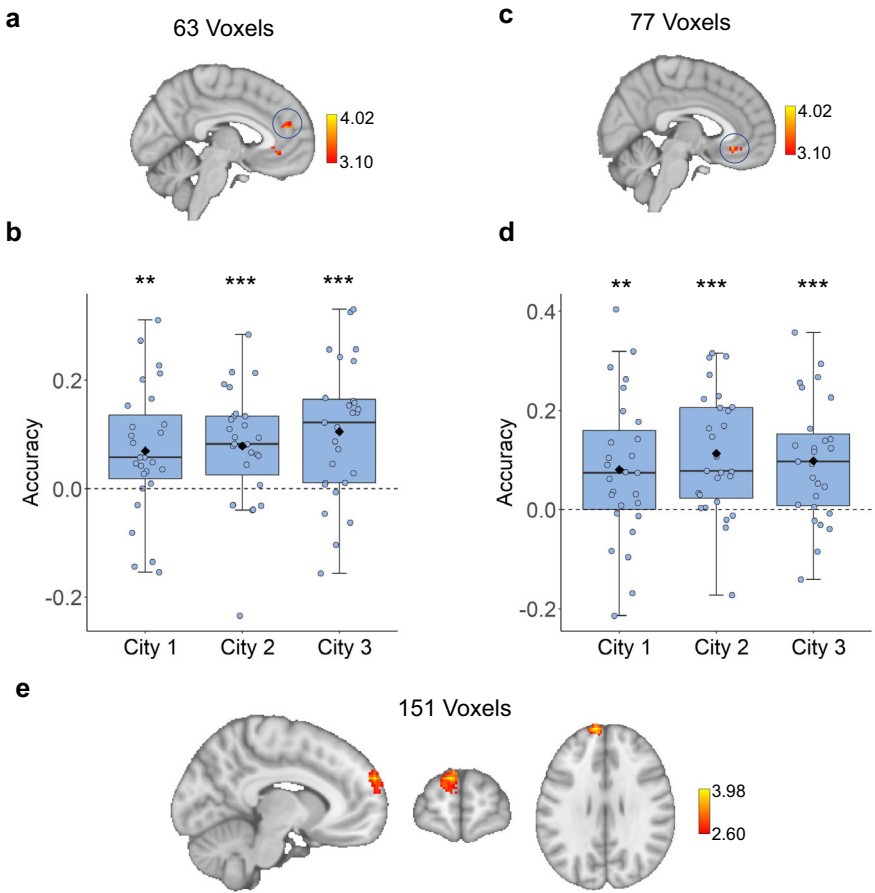

**Fig. 4 Representation of schematic spatial layout information across environments in medial PFC. a** The searchlight leave-one-city-out SVR classification revealed a more superior cluster (MNI: −4, 50, 18, Z = 3.68) in mPFC whose spatial distances generalized from training on two cities to a third one (a random-effects model was used for group analyses within the mPFC mask using a cluster-forming threshold of Z > 3.10, with p < 0.05 (corrected for family-wise error rate, using random field theory). **b** The SVR performance for each city in superior mPFC revealed above chance performance on cities 1, 2, and 3 (City 1: p = 0.005, City 2: p < 0.001, City 3: p < 0.001, two-tailed one-sample *t*-test with FDR correction, compared to chance level of 0). **c** The searchlight leave-one-city-out SVR classification revealed a more ventral cluster (MNI: −10, 34, −4, Z = 4.02) in mPFC whose spatial distances generalize from training on two cities to a third one (a random-effects model was used for group analyses across within the mPFC mask using a cluster-forming threshold of Z > 3.10, with p < 0.05 (corrected for family-wise error rate, using random field theory). **d** The SVR performance for each city in ventral mPFC revealed above chance performance on cities 1, 2, and 3 (City 1: p = 0.008, City 2: p < 0.001, City 3: p < 0.001, two-tailed one-sample *t*-test with FDR correction compared to chance level of 0). **e** The searchlight revealed a significant cluster located in medial frontal pole (MNI: −8, 62, 28, Z = 3.98) whose between-city PS for the same locations was higher than for different locations (a random-effects model was used for group analyses within the mPFC mask using a cluster-forming threshold of Z > 2.60, with p < 0.05 (corrected for family-wise error rate, using random field theory). Notes: Boxplots are centered on the median, boxes extend to first and third quartiles, whiskers extend to 1.5 times the interquartile range or minima/maxima in the absence of outliers. Each unfilled dot represents data from an individual subject. Each black solid diamond represents the mean of the group. All data reflect n = 27 independent participants. mPFC medial prefrontal cortex, SUB subiculum. **p < 0.01, ***p < 0.001. Source data are provided in the Source Data file.

p = 0.008, CI = [0.081 ± 0.055], Cohen's d = 0.555; City 2: t(26 = 4.800, p < 0.001, CI = [0.113 ± 0.046], Cohen's d = 0.924; City 3: t(26) = 4.190, p < 0.001, CI = [0.098 ± 0.144], Cohen's d = 0.806; chance level = 0, two-tailed, FDR corrected, Fig. 4d). Note that these two clusters were the only two that survived whole brain correction, suggesting schema-like codes for shared spatial positions within mPFC.

As a control analysis, we tested whether this effect was driven by the shared stores. In this case, we would expect that City 2 (with the most shared stores, Supplementary Fig. 5c, see also Supplementary Note 6) would show the highest classification accuracy, which we did not find (t(26) > 1.138, Ps > 0.266, Fig. 4b and d see also Supplementary Note 6). We also examined whether the spatial schema effects were present for temporal durations estimates using a searchlight-based SVM classification analysis (see "Methods"); however, no clusters exceeded chance levels. In

addition to the SVR classification analysis, we also tested whether MPS might reveal quantitatively similar relationships in neural patterns for shared locations. Therefore, we tested the spatial schema hypothesis by performing a searchlight-based MPS throughout the whole brain (see "Methods"), predicting that if the mPFC contained shared layout information across cities, the same location between-city PS would be higher than different location between-city PS. As predicted, this analysis revealed a significant cluster located in medial frontal pole (MNI: −8, 62, 28, Z = 3.98, Fig. 4e and Supplementary Fig. 6d; note that we applied a lower threshed (Z = 2.6) and a small volume correction within mPFC mask). Together, these findings suggest that mPFC contains representations with general spatial distance codes.

Next, we examined whether hippocampus also contained such spatial schematic representations by performing the same SVR classification analysis and MPS in each of the hippocampal

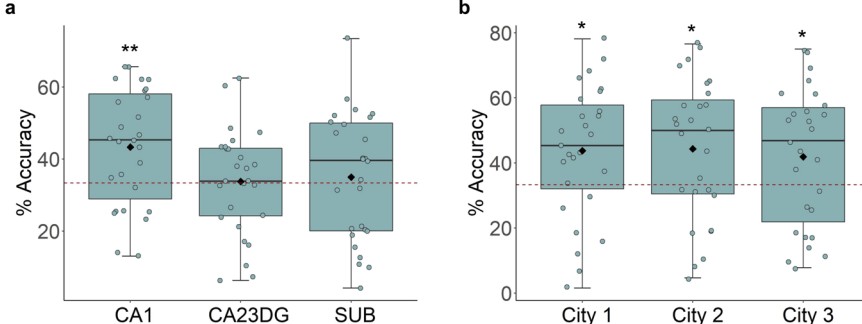

**Fig. 5 Environment-specific representations in CA1. a** The SVM classifier revealed overall classification accuracy in CA1 (across all cities) was well above chance (t(26) = 3.222, p = 0.009, two-tailed one-sample t-test compared to chance level = 33.33% which survived FDR correction). **b** Testing each city's classifier performance against chance in CA1 revealed that the classifier performed above chance on all cities (City 1: p = 0.020, City 2: p = 0.020, City 3: p = 0.045, two-tailed one-sample t-test compared to chance level = 33.33% with FDR correction). Notes: Boxplots are centered on the median, boxes extend to first and third quartiles, whiskers extend to 1.5 times the interquartile range or minima/maxima in the absence of outliers. Each unfilled dot represents data from an individual subject. Each black solid diamond represents the mean of the group. All data reflect n = 27 independent participants. mSUB subiculum. *p < 0.05, **p< 0.01. Source data are provided in the Source Data file.

subfields. No ROIs within the hippocampus showed above chance level classification for spatial distance estimates ($P$s > 0.631, Supplementary Fig. 6a) or for temporal interval estimates ($P$s > 0.134, Supplementary Fig. 6b). In addition, no hippocampal ROIs showed higher between-city PS for the same locations than different locations (Supplementary Fig. 6c, $P$s > 0.237), suggesting that the hippocampus did not contain such schematic layout codes. Because a previous study from our lab[15] and rodent studies using single-neuron recordings have identified environmental-specific codes in the hippocampus[65–67], we also tested whether the hippocampus might instead contain such differentiated environment-specific codes, despite significant overlap between the different cities in our task (Fig. 1a). To address this prediction, we performed a SVM classification analysis to classify the three cities based on all hippocampal ROIs (see "Methods") as participants retrieved information about spatial distances and temporal intervals (Fig. 1e). This approach allowed us to identify hippocampal regions where voxel patterns carried city-specific information (see "Methods").

The pattern classifier revealed classification rates (across all cities) well above chance level in CA1 (mean = 43.33%, SD = 16.12%, t(26) = 3.222, p = 0.009, CI = [0.433 ± 0.061], Cohen's d = 2.688, chance level = 33.33%, FDR corrected, Fig. 5a; all cities against chance, City 1: t(26) = 2.662, p = 0.020, CI = [0.438 ± 0.077], Cohen's d = 2.151; City 2: t(26) = 2.648, p = 0.020, CI = [0.443 ± 0.082], Cohen's d = 2.054; City 3: t(26) = 2.108, p = 0.045, CI = [0.419 ± 0.080], Cohen's d = 1.984; two-tailed, FDR corrected, Fig. 5b). Importantly, there were no significant differences either in memory performance or reaction time (RT) across the three cities (see Supplementary Note 1), suggesting that classification accuracy was not confounded by a difference in behavioral performance across the three cities (please see Supplementary Note 1 for detailed behavioral results and analyses). These findings replicate our previous findings of environment-specific codes within the hippocampus but extend them to suggest that even cities sharing significant environmental overlap showed distinct neural representations within CA1[15]. These findings provide additional support for the idea of environment-specific codes within the hippocampus.

## Discussion

Here, we asked how aspects of spatial memory retrieval are implemented in the human brain and how such spatial memory representations are differentiated when they involve overlapping features. We found that the hippocampus could reinstate non-target but unique elements within the same environment during retrieval, providing empirical evidence for "holistic retrieval" of spatial memory. Then, we further investigated the neural mechanisms for resolving interference when dealing with shared features across different environments. On one hand, our findings suggested that the hippocampus disambiguated the representation of overlapping elements across environments via pattern separation[12–14] and repulsion mechanisms[24–28]. On the other hand, neural activity in the lateral PFC contributed to holistic spatial retrieval by potentially accentuating hippocampal representations of landmarks unique to an environment rather than shared or unrelated ones. In addition, abstract spatial schema, which our findings suggested were specific to the medial PFC, played a role in providing the common spatial structure across the three environments.

Previous work has demonstrated aspects of holistic retrieval as part of event (episodic) memory, i.e., that multiple elements involved in an episode can be retrieved incidentally[2,3,5,6]. Whether spatial memory retrieval, however, exhibits the same property neurally has not been tested previously, with holistic retrieval in particular considered a critical property of cognitive maps[1,68]. In this study, we employed store templates, based on a post-task scan, to determine the neural patterns associated with each of the different stores participants studied. The neural templates of stores can be seen as the memories obtained from learning the three different environments. Based on the idea of environment-specific codes, the neural representational distances of stores within the same city would be closer to each other than the neural representational distances of stores across cities. Guided by this hypothesis, we found that the multivoxel activity patterns of incidental stores within that city showed higher reinstatement than those unique to another city, providing strong evidence for holistic retrieval within the hippocampus. In addition, our findings showed that holistic reinstatement is task-independent, though this effect was somewhat weaker when only one task was considered (see Supplementary Note 2). This suggests, that regardless of whether participants retrieved spatial distances or temporal durations, whenever they accessed the environment in our experiment, they retrieved environment-specific codes. Together, our results provide important evidence for holistic retrieval in human spatial memory, a cornerstone of the idea of a "cognitive map" and one means by which spatial information is stored collectively.

As we noted in the introduction, however, one issue with holistic retrieval is that it does not provide an obvious means for

distinguishing shared elements across environments. Accumulating evidence suggests that pattern separation within the hippocampus may serve as an important means for distinguishing similar stimuli[12–14], with our findings suggesting such a mechanism, at least as detected using MVPA and high-resolution fMRI, could play a partial role in distinguishing shared elements across cities. In the current study, we correlated the trials involving at least one unique store with trials involving stores unique to another environment by excluding pairs that had overlapping stores to create a clearer and meaningful baseline. Therefore, the comparison of the similarity of shared city trials between different environments to the similarity of trials unique to an environment can be used to access the extent of differentiation among shared city trials. In CA2/3/DG, we found neither the similarity of two nor three shared city trials was significantly different from baseline, likely reflecting orthogonal memory representations[13,15–19,69]. Such distinct representations of shared city trials are likely important to discriminating elements within similar environments[70]. Particularly in our experimental design, the shared city trials involved identical stores and were shared between two or three cities. Therefore, reduction in the distributed patterns of neural activity in CA2/3/DG may be important to forming distinct codes for these competing trials to reduce memory interference and allow successful retrieval of environmental information.

At the same time, we found evidence for a different mechanism within CA1. The between-city PS of three shared city trials in hippocampus was significantly lower than that for unique and two shared city trials, representing a significant reduction in overlapping memories, which is called "repulsion"[24–28]. Past studies have shown pattern separation mechanisms in the hippocampus using single-neuron activity[16,20], multi-neuron activity patterns[21], LFP activity patterns[22], and fMRI activity pattern[15], and while only past fMRI studies have shown repulsion[24,71], we note that theoretically, such a mechanisms could represent a viable, although untested, manner by which single-neuron activity and LFPs could also process overlapping information. Such reverse representations have been reported previously for overlapping routes during virtual navigation using fMRI[24] although, to the best of our knowledge, not in any previous rodent single-neuron studies or fMRI studies requiring environment-specific retrieval. Here, by using high-resolution fMRI, we provide evidence for a functional role of "repulsion" in CA1 for separating overlapping information during spatial retrieval

At first blush, this result seems at odds with some theoretical models and animal studies suggesting that CA1 is less sensitive to small environmental changes compared to CA2/3/DG and may play a greater role, in some instances, in pattern completion for previous environments rather than remapping[17,72–74]. Because CA1 receives direct input from subfields CA3 and DG, however, it is possible that CA1 can serve to further discriminate shared information as part of interference reduction. Such a finding is also consistent with theoretical models suggesting that CA1 may represent changes in input in a linear fashion[74], by which its representational space/range is large enough to allow "reverse" representations to occur–even past the point of orthogonalization. According to this account, positive, zero, and negative MPS correlations are simply different degrees of similarity existing along a continuous spectrum of pattern separation/completion. Such a mechanism would allow greater amplification of differences among environmental representations when they are identical, such as occurred in our paradigm, to achieve precise spatial memory for each city. Indeed, this result was most prominent for three shared city trials while at the same time, in CA1, pattern similarity for two shared city trials was comparable to unique-city trials, suggesting a pattern separation effect. Past

results suggest that repulsion mechanisms may necessitate significant learning before they arise[24], and it is possible, in an analogous way, that three city-shared trials represented the type of overlearning requiring such repulsion. Future studies, however, will need to better define the conditions under which repulsion and pattern separation might operate. Finally, given the lack of a direct mapping between hippocampal BOLD fMRI and single-neuron/LFP activity recorded at single electrodes in the hippocampus[75,76], it is unclear exactly how the fMRI findings here related to pattern separation and repulsion in particular might relate to the activity of place cells. While there is evidence supporting a correspondence between multivariate patterns at the level of fMRI BOLD activity and populations of single neurons/LFPs[77–81], whether a population of single neural/LFP codes exhibit the same reversed codes is still need of direct testing. To more directly address this issue, we performed simulations of how large groups of neurons displaying distributed codes might relate to distributed patterns of voxel-based activity using fMRI. Our simulations support the idea that, under some situations, both "separation" and "repulsion" mechanisms at the level of single neurons / LFPs, provided they are sufficiently distributed, can be detected with MVPA methods at the level of changes in patterns across voxels (Supplementary Note 7). Overall, these results support adaptive and flexible representational mechanisms within the hippocampus as important to mediating memory interference[20,25,26,82].

Is it possible that the evidence we obtained for repulsion could arise from other factors? Because the three shared city trials were the trials composed by stores shared across three cities, could our effects have been driven by repetition suppression? A control analysis, however, in which we compared activation levels between unique city trials and three shared city trials did not reveal any differences (see Supplementary Note 3). Because stores shared across the three different cities were seen more often during encoding, could it be that attention demands drove our effects by altering the noise[83] of neuronal representations for stores unique to a city? We did not observe any significant differences in reaction time or accuracy between the unique city trials and three shared city trials, indicating no observable significant attention differences between the two conditions (see Supplemental Note 1 for a full presentation of the behavioral results). Finally, as we discussed above, CA1 showed reinstatement of unpresented unique stores specific to an environment during spatial memory retrieval. In this case, when retrieving the three shared city trials, the unique stores of the current city might reinstate incidentally, adding unique information to the representation of the three shared city trials, therefore increasing dissimilarity between those trials. However, if the low similarity of three shared city trials were fully attributed to the reinstatement of unique stores, the representations of three shared city trials should not be lower than unique city store trials (i.e., orthogonalization). Therefore, this account fails to explain why the representations of the three shared city trials were past the point of orthogonalization. Instead, we favor an explanation in which trials involving stores shared across all three cities involved a completely different neural representation, which would be better supported by mechanisms like repulsion compared to pattern separation.

Despite the strong evidence that our findings provided for a role of the hippocampus in representing the different environments in our task via pattern separation and repulsion, we also found evidence that the PFC played important role in our task. Previous research suggests that the lateral PFC is believed to be involved in the control processes such as attention, selection, updating, and maintenance[84,85]. Specially, past studies have suggested an important role for the lateral PFC important for

planning[86] and detecting novelty[87] during navigation. These control processes might promote interference resolution by strengthening task-relevant hippocampal representations[62,63]. Supporting this idea, our results provide evidence for a role for PFC in facilitating spatial holistic retrieval: we found a positive correlation between ventrolateral PFC (i.e., LIFG) activity and holistic retrieval in the hippocampus, which was only true for landmarks unique to an environment (Fig. 2c, d). Our findings, therefore, suggest a possible framework for spatial holistic retrieval: the hippocampus is not the only region responsible for incidental reinstatement of unique information within an environment, and interactions with the lateral PFC could provide a mechanistic basis for accentuating the fidelity of the reinstated stores as part of mediating interference during successful retrieval. Neurostimulation or lesion studies would be necessary, however, to determine the directionality of this potential influence. Another possibility, although less likely given the suggested role of PFC in cognitive control, is that hippocampal subfields modulate schema-like representations in the PFC or both regions interact in some to-be-defined manner during holistic retrieval.

To successfully complete both the spatial distance and temporal interval judgments required during retrieval, participants would also need to learn the locations of each of the stores. Although store identity changed between the different environments, their locations remained constant (Fig. 1a, c), and therefore participants likely abstracted a representation of the locations of the stores common to all cities. This allowed us to address whether such a spatial "schema" was represented in the hippocampus or in PFC, with past studies supporting both possibilities[40–47,88–90]. A recent primate study found direct evidence that macaque hippocampal cells abstract a generic spatial schema from repeated experiences of one environment, regardless of the surface features facilitating learning in a novel environment[40]. In contrast, another recent non-human primate study did not find evidence for a common representation of location within the hippocampus[91], consistent with the idea that such schema representations may be stored elsewhere. Although our study did not involve single-neuron recordings, and therefore may have involved different forms of neural-related information coding schemes, we did not find evidence for spatial schema within the hippocampus. Specifically, our results from a leave-one-city-out classification analysis suggested that spatial distances learned from two cities did not generalize to the third new city in hippocampus and results from a MPS analysis also suggested that the between-city PS for the same locations was not higher than different locations in hippocampus. On the contrary, the same leave-one-city-out classification and MPS analyses revealed evidence for spatial schema within mPFC.

The leave-one-city out SVR classification analysis places more emphasis on decoding spatial distance information while the MPS approach focused on the same location information across three cities. While both two approaches revealed different aspects of spatial schema representations, the SVR findings may be better positioned than the MPS findings to allow inference about spatial schema because the SVR analysis explicitly utilized distance measures. Therefore, the present findings are consistent with previous studies in which mPFC is involved in schema congruency effects during encoding, post-learning consolidation, and retrieval[92], and also in accordance with the SLIMM (schema-linked interactions between medial prefrontal and medial temporal regions) model[43] which assume reciprocal mPFC-hippocampus coupling[93,94]. Taken together, our findings demonstrate that the hippocampus and mPFC play specific but important roles in spatial memory retrieval in our task. We found that the hippocampus was more involved in environment-specific rather than generic spatial layout representation while the mPFC

played a critical role in representing common elements across environments in the form of spatial schema.

In summary, the findings of the study provide support for holistic retrieval in spatial memory retrieval, leading us to propose a framework involving multiple interacting brain regions to explain how this might occur. Importantly, we found that two brain regions in particular interact with each other to accomplish successful spatial memory retrieval: the hippocampus is involved in detailed and differentiated memory representation to resolve interference when encountering multiple similar environments while the mPFC is involved in representing a common spatial schema across different learning environments to facilitate spatial memory retrieval. In this way, both areas are central to spatial memory retrieval, but play different roles based on the task demands.

## Methods

**Participants**. A total of 32 right-handed participants were recruited from the Tucson community and were compensated for their time. Four participants were excluded from the analysis due to excessive movement (>1 voxel), and one participant was excluded due to an incidental finding. Therefore, the final sample size was comprised of 27 participants (17 females, mean age: 22.52 years, range: 18–35 years). All participants had normal or normal-to-corrected vision and normal color perception. Based on self-report, all participants were screened to ensure they had no neurological conditions. The study was approved by the Institutional Review Board at the University of Arizona and written Informed consent was obtained from each participant prior to the experiment.

**Materials**. The experiment consisted of two sessions: an encoding session outside the scanner and a retrieval session inside the scanner. Three different virtual environments were created using Unity3D (https://unity3d.com). The three different cities contained stores arranged in a circle, and each consisted of six different stores located on the edge of the circle. One store ("Camera Store", which was consistent across three environments) was in the center of the circle (Fig. 1a), allowing us to manipulate temporal duration while holding spatial distance constant. All three cities had the same basic layout (Fig. 1c), including the same ground and wall textures; thus, cities only varied in terms of what stores were shared or distinct across cities (Fig. 1a).

The degree of similarity between cities was proportionate to the number of overlapping edge stores between cities (we did not take the center "Camera Store" into consideration, because the center store was the same across the three cities, Fig. 1b). Specifically, Cities 1 and 2 were the same except the "Store 4" in City 1 was changed to "Store 7" in City 2. Cities 2 and 3 were the same except the "Store 2" and "Store 6" in City 2 were changed to "Store 8" and "Store 9" in City 3, respectively. Therefore, Cities 1 and 2 had 5 overlapping stores (i.e., "Store 1", "Store 2", "Store 3", "Store 5" and "Store 6"); Cities 2 and 3 had four overlapping stores (i.e., "Store 1", "Store 3", "Store 5" and "Store 7"); Cities 1 and 3 had three overlapping stores (i.e., "Store 1", "Store 3" and "Store 5"). All the three cities also had three overlapping stores (i.e., "Store 1", "Store 3" and "Store 5", Fig. 1b).

For the temporal interval task, participants encoded two different durations: 8 s (i.e., "Store 3", "Store 5", "Store 6" and "Store 9") and 16 s (i.e., "Store 1", "Store 2"," Store 8", "Store 4" and "Store 7", Fig. 1a).

## Experimental procedures

*Prescan encoding*. During the encoding session, the participants were trained to perform two tasks involving navigating each city: a spatial distance task and temporal interval task (Fig. 1d). Participants were instructed that there would be three cities, with some stores shared across cities (see "Methods"), and that there would be a spatial distance/time interval retrieval task to test whether they could successfully learn this information in each city. At the beginning of each round of the spatial distance task, participants were placed at the center of the city and viewed videos of travel from the center store to peripheral stores in a randomized order. All the participants were asked to learn as much as they could about store locations during traversals. Similarly, in the time interval task, participants were asked to learn the time interval between the center and each store. We consider differences between the distance and interval tasks in a separate paper, with our focus here on retrieval of the different spatial environments, which would be common to both tasks. To ensure that participants did not merely use a counting strategy to encode time interval information, we added a distractor task ("math problem") during the route from the center of the city to each store. A math problem would pop up in a pseudorandom position during the route to each store. Participants were asked to focus on processing the time interval while at the same time correctly answering the arithmetic question by the time they arrived at the store. This ensured that participants were not using a counting task to encode temporal intervals. Participants needed to determine their answer and submit it when they reached the store. Furthermore, the travel speed from the center to each

store was not constant but variable to avoid the possibility that participants could merely take advantage of the speed differences to discriminate time intervals.

The encoding process repeated as many times as the participants needed in order to learn spatial distances and temporal intervals before they moved on to the next city. The learning order of type of task (space/time) was randomized across all participants. After participants learned all three cities for one type of task (e.g., spatial distance), they then learned three cities for the other task (e.g., temporal interval task). Before starting the main encoding task, participants also performed a practice session in which they visited three additional stores in a virtual city to familiarize themselves with the main task.

After the spatial and time encoding task, we tested participants' memory for each of the three cities for both the spatial distance and temporal interval by performing a shorter version of the memory retrieval task they would experience in the scanner. The short version retrieval task was the same as the main fMRI memory retrieval task (see fMRI memory retrieval task, Fig. 1e) but only included 12 trials of spatial distance questions (not used in fMRI task) and 5 trials of temporal interval questions (used in fMRI task) for each city. If a subject failed to reach the memory accuracy criterion (i.e., 80%), they re-learned and were re-tested on their memory for all three cities.

**fMRI memory retrieval task.** The fMRI retrieval session consisted of six consecutive spatial runs and six consecutive temporal runs (two per city), each including 15 trials and lasting 4 min and 40 s pertaining to a single city and a single task. The order of retrieval runs (spatial or temporal) across participants was fully counterbalanced and was pseudo randomized with rules dictating that no city could be tested twice in a row and that each city must be tested once before a city could be repeated. The spatial and temporal retrieval probes were rendered identically during retrieval. Before starting each retrieval run, text and verbal instructions reminded participants of which city and which type of task they would be retrieving next, followed by a 7.77 s (3 TRs) refresher picture which included all the stores of that city. There were not shown the actual layout of the city just pictures of stores to cue the correct city.

A slow event-related design (18.13 s for each trial) was used in this study to better characterize the activation pattern for each trial (Fig. 1e). During spatial distance blocks, participants were instructed to retrieve the spatial distance by making judgments of the relative distances of stores in that city. For each trial, participants saw three stores on the screen for 9 s, with one store on the top and two below (Fig. 1e). Participants were asked to compare which of the two bottom stores was closer to the upper reference store and indicate their choice by pressing the corresponding key on an MR-compatible button box. A "one" response indicated that the lower-left store was closer to the top store, a "two" indicated the lower right store, and a "three" indicated that the distance from the two bottom stores to the reference store was equal. For temporal trials, the store in the center of the city ("Camera Store") always appeared on the top of the screen, and two peripheral stores appeared on the bottom. Participants were instructed to judge which of two intervals between the center (top) and bottom stores was shorter. Once participants pressed the button within 9 s, a black outline would appear to indicate that they have completed the current question, while these three stores would stay on the screen until 9 s finished. Next, participants performed an active baseline task for 7.77 s, in which they pressed "one" for the appearance of an "X", and "two" for the appearance of an "O"[95]. A self-paced procedure was used to make this task engaging; each letter appeared 0.2 s after the response.

One hundred and eighty trials were presented in 12 runs, with half as spatial runs and half as temporal runs. One hundred and eight of these trials (60% of total trials) presented "unequal" comparisons in which the two bottom stores were an unequal spatial or temporal distance from the reference store. Seventy-two of these trials (40% of total trials) presented "equal" comparisons in which the two bottom stores shared an equal spatial or temporal distance from the reference store.

**fMRI localizer task.** After the retrieval task, participants were asked to complete a localizer task involving a vowel counting task, which included two runs, each containing 18 trials (~6 min). This task served as a localizer task to allow the creation of multivariate pattern templates for each of six stores (i.e., "Store 2", "Store 4", "Store 6", "Store 7", "Store 8", "Store 9", see more details in tSNR based fMRI MPS). The structure of each trial in the vowel counting task was the same as in the retrieval task (Fig. 1e). Here, participants were asked to count the number of vowels (i.e., "A", "E", "I", "O", "U"; "Y" did not count) in each of the three store names and then select which of the two bottom stores had the closest number of vowels compared to the store on the top. Participants were asked to perform both vowel counting and X/O judgment task as accurately and quickly as possible.

To allow us to build store "templates", each triad of stores included one old store which was presented in the retrieval task with two new stores for vowel counting. These new stores were randomly selected from 24 unstudied stores. The purpose of the new stores was to allow us to identify unique activation patterns associated with a specific old store while at the same time allowing us to keep the trial structure the same as during the spatial retrieval questions. The positions of the old stores in each triad were counterbalanced such that all position arrangements for each old store were presented (i.e., old store-new store A-new store B, new store C-old store-new store D, new store E-new store F-old store).

Thus, each old store was repeated three times in a different position of the triad within a run with an inter-repetition interval ranging from 2 to 12 trials.

**MRI image data acquisition.** All participants were tested immediately following encoding in the Siemens 32-Channel 3 T "Skyra" scanner, located in the University of Arizona. Visual stimuli were projected onto a screen behind the scanner, which was made visible to the participant through a mirror attached to the head coil. Stimuli and responses were presented and recorded by PsychoPy (https://www.psychopy.org) on a Windows laptop. High-resolution functional images were acquired using a simultaneous multi-slice whole-brain echo planar imaging (EPI) sequence (interleaved acquisition, TR = 2590 ms, TE = 30 ms, flip angle = 82 degree, field of view (FOV) = 234 mm, matrix = 128 × 128, slice thickness = 1.8 mm, slices = 84, slice acceleration factor = 3, phase encoding direction = right to left, bandwidth = 1562 Hz/pixel), adapted from a previous study[96]. High-resolution structural images were obtained using a 3D, T1-weighted MPRAGE (1 mm³ isotropic) sequence acquired for the whole brain (FOV = 256 mm, matrix = 256 × 256, slice thickness = 1 mm, TR = 2100 ms, TE = 2.33 ms, flip angle = 12 degree, bandwidth = 190 Hz/pixel). High-resolution anatomical images of the hippocampus and surrounding cortex were acquired with a T2-weighted turbo-spin echo (TSE) anatomical sequence (FOV = 200 mm × 200 mm, matrix = 448 ×;448, TR = 4200.0 ms, TE = 93.0 ms, flip angle = 139 degree, slice thickness = 1.8 mm, 28 slices, bandwidth = 199 Hz/pixel). Sequences were acquired perpendicular to the long axis of the hippocampus. An additional coplanar matched-bandwidth high-resolution gradient-echo EPI sequence (TR = 6120 ms, TE = 39 ms, slices = 84, FOV = 245 mm, matrix = 128 × 128, flip angle = 90 degree, bandwidth = 1446 Hz/pixel) was acquired to aid in registration of the EPI sequence to the high-resolution structural images. B0-field maps were acquired immediately with a gradient recalled echo sequence (TR = 888.0 ms, TE1 = 4.92 ms,TE2 = 7.38 ms, flip angle = 90 degree, FOV = 256 mm, slice thickness = 3 mm, slices = 84) following the coplanar matched-bandwidth sequence to correct for inhomogeneities of the magnetic field[97]. This sequence covered the whole brain, allowing us to correct field distortions for the entire EPI sequence.

**fMRI data preprocessing.** Image preprocessing was performed by using FEAT (FMRI Expert Analysis Tool), version 6.00, implemented in FSL (part of the FSL package; http://www.fmrib.ox.ac.uk/fsl). The first seven images were automatically discarded from each run by the scanner to allow for scanner to equilibrate. We additional discarded 3 volumes in which the refresher picture was present before the retrieval task started. The EPI images were first corrected for geometric distortion using participants' field maps[97,98] and underwent motion-correction, temporal filtering (nonlinear high-pass filter with a 100 s cutoff), and slice-timing correction. Six motion parameters were added as confound variables to the model. Residual outlier timepoints were identified using FSL's motion outlier detection program and integrated as additional confound variables in the first-level general linear model (GLM) analysis. No spatial smoothing was applied for single-trial estimation (see below). All functional images were linearly registered to individual-subject T1 MPRAGE structural volumes in a two-step process via a coplanar matched-bandwidth sequence described above using FLIRT. Registration from structural images to the standard MNI-152 template was further refined using FNIRT nonlinear registration for higher-level group analysis when needed (see below).

**Single-trial response estimates.** The GLMs were performed separately to estimate the activation pattern for each of 180 retrieval trials and 36 localizer trials. In this single-trial model, a Least Square–Separate (LS-S) approach was used, in which the trial of interest was modeled as one regressor, with all other trials modeled as a separate regressor[99]. Specifically, each single-trial GLM included five regressors: (1) the trial of interest; (2) all other trials; (3) black outline stage; (4) fixation; (5) all incorrect trials within the active baseline task. Each event was modeled at the time of stimulus onset and convolved with a canonical hemodynamic response function (double gamma), whereas the correct baseline trials (X/O judgment task) were not coded and thus were treated as an implicit baseline. To control for the effects of head motion, six motion parameters were included in the GLM model as a covariate. The t-map for each trial was used for multivariate pattern similarity analysis (MPS) and SVR classification analysis to increase the reliability by normalizing for noise[100].

**Run-based response estimates for SVM classification analysis.** The GLMs were performed separately to estimate the activation pattern for each retrieval run. Each single-run GLM included 6 regressors: (1) the remembered trials; (2) forgotten trials; (3) missed trials; (4) black outline stage; (5) fixation; (6) all incorrect trials within the active baseline task. Each event was modeled at the time of stimulus onset and convolved with a canonical hemodynamic response function (double gamma), whereas the correct baseline trials (X/O judgment task) were not coded and thus were treated as an implicit baseline. To control for the effects of head motion, six motion parameters were included in the GLM model as a covariate. This resulted in 4 run-based data points per city per subject. The run-based

t-map has greater reliability[101] and could be used for SVM classification analysis to increase accuracy[102] and power[103].

**Subfield demarcation and ROIs.** Automatic hippocampal subfield segmentation software (ASHS)[104,105] was used to segment the subregions of the MTL based on each participant's high-resolution T2-weighted MRI image. The MTL was segmented into CA1, CA2/3, DG, and subiculum (SUB), perirhinal cortex (PRC) and entorhinal cortex (ERC) and parahippocampus cortex (PHC). We combined the CA2/3 and DG subfields as finer distinctions cannot be made at the acquired resolution. Single-trial t-map were then obtained within those 6 ROIs (CA1, CA2/3/DG, SUB, ERC, PRC, PHC) for each subject for further MPS and classification analysis. Following a previous study[43], the medial PFC mask was defined as a set of three regions within the Brodmann areas (BA) 10, 11, and 32.

**Temporal signal-to-noise ratio (tSNR).** We adopted voxel-wise tSNR to define the fMRI time series stability[106]. Specifically, for each MTL ROI, we obtained the voxel-wise tSNR of localizer task by calculating the mean of each voxel's time series divided by its standard deviation. Then the voxels in each ROI could be ranked from high to low by the intensity tSNR and could be further divided into eight portions by different levels of percentile (i.e., 10th, 20th, 30th, 40th, 50th, 60th, 70th, 80th). For example, "10th percentile of tSNR" means that voxels with tSNR less than the bottom 10% of tSNRs in the ROI were removed from the analysis. Therefore, by applying different percentile tSNR as the threshold of t-stat, we could exclude different levels of influence of spurious voxels in the MPS.

**tSNR based fMRI multivariate pattern similarity analysis (MPS).** *Multivariate patterns of stores.* In the localizer task, six studied stores (i.e., "Store 2", "Store 4", "Store 6", "Store 7", "Store 8", "Store 9") were repeated 6 times (3 times per run, see Procedures). Based on the specificity, these six stores could be classified into two categories: unique city stores and stores shared across cities. For example, "Store 4" (only belongs to City 1),"Store 8" (only belongs to City 3), and"Store 9" (only belongs to city 3) are unique city stores because they were only presented in one city, while "Store 2" (belongs to City 1 and City 2), "Store 6" (belongs to City 1 and City 2) and "Store 7" (belongs to City 2 and 3) are stores shared across cities, because they were presented in two cities. Then, we constructed a multivariate pattern template for each of the six studied stores that were presented in the localizer task by averaging the activation patterns (i.e., single-trial t-maps) across six repetitions of a given store. The template of each store could provide a neural measure for a memory trace of each store during memory retrieval. Because the vowel counting task, which occurred at the end of the fMRI session, did not involve spatial retrieval and occurred after participants had retrieved information from all three environments, it is unlikely that the templates contained any environment-specific information and therefore could provide indices to store identity.

We then applied MPS by measuring the similarity of activation patterns between each of the six store templates and each remembered trial in both retrieval tasks (spatial and temporal) based on the different thresholds of tSNR in each hippocampal subfield. We followed the approach of Power et al.[107] and censored TRs with a framewise displacement >0.5 mm. Specifically, to quantify unique store within-city pattern similarity (PS), pairwise Pearson correlation coefficients were calculated by correlating each unique city store template with the activity patterns evoked by correctly retrieved trials that did not include the given store within that specific city. For example, to determine whether participants retrieved a store not contained in a retrieval triad, we correlated the template of "Store 4" (belongs to City 1) from the control task with any correctly retrieved trial which did not include "Store 4" within City 1). Similarly, the unique store between-city PS was the correlation between each unique city store template and the activity pattern elicited by the remembered trials that did not belong to the current city. For example, to determine whether participants retrieved a store not contained in a retrieval triad, we correlated the template of "Store 4" (belongs to City 1 only) from the localizer task with any remembered trial of City 2 and City 3. Finally, the shared store within-city PS was the correlation between each store shared across city template and the activity pattern of each remembered trial that did not include the current store within all the shared cities. For example, to determine whether participants retrieved a store not contained in a retrieval triad, we correlated the template of "Store 2" (belongs to City 1 and City 2) from the localizer task with the activity pattern of remembered of trials without "Store 2" within City 1 and City 2. Since correlations are inherently a pairwise comparison, many correlations were performed and then averaged together for a metric of within-condition similarity. The resulting correlation coefficients were transformed into Fisher's z-scores and then input into further group analyses.

To examine the repulsion hypothesis, we also applied the tSNR based MPS. All the remembered trials of the retrieval task could be divided into three conditions: unique city trials (i.e., the trial that could only be attributed to one city, for example, the triads "Store 1-Store 4-Store 5" was only attributable to City 1 and the triads "Store 8-Store 9-Store 3" was only attributable to City 3, Fig. 4c, top panel), two shared city trials (i.e., the trial could be attributed to two possible cities, for example, the triads "Store 2-Store 1-Store 6" and "Store 2-Store 6-Store 1" could only be attributed to City 1 and City 2 but not be attributed to city 3, Fig. 4c, middle panel) and three shared city trials (i.e., the trial could be attributed to three

possible cities, for example, the triads "Store 1-Store 3-Store 5" and "Store 1-Store 5-Store 3" could be attributed to City 1, City 2 and City3, Fig. 4c, bottom panel). We calculated the between-city pattern similarity of the independent trials that corresponded to a specific condition (unique city trials, two shared city trials and three shared city trials) separately for spatial and temporal retrieval tasks. Note that all MPS analyses involved correlating between different runs of retrieval, thus avoiding temporal autocorrelations artificially inflating or biasing results.

In addition, we also calculated between-city pattern similarity of trials for a specific condition across spatial and temporal tasks to utilize as many as possible correlations to obtain stable metric within a condition. Note, for those pairs which were included in the between-city PS of unique-city trials calculation, we excluded the unique-city pairs that have overlapping stores, for example, the triad "Store 1-Store 4-Store 6 and triad "Store 1-Store 8-Store 7", to make the representation of each triad as distinct from each other as possible. Because three shared city trials that involved in the between-city PS calculation are perceptual identical, for example, the correlation between triad "Store 1-Store 3-Store 5" and triad "Store 1-Store 5-Store 3". We also matched the two shared city trial pairs by selecting the pairs that had the same stores, for example, we only calculated the correlation between triad "Store 1-Store 2-Store 6" and "Store 1-Store 6-Store 2", we did not calculate the correlation between triad "Store 1- Store 2-Store 6" and triad "Store 1-Store 3-Store 6". The resulting correlation coefficients were transformed into Fisher's z-scores and then input into further group analysis.

**Searchlight-based MPS.** To examine the shared spatial layout information across three cities, we applied the MPS throughout the whole brain using searchlight approach[60]. For each voxel, signals (i.e., single-trial t-maps) were extracted from a cubic ROI containing 343 surrounding voxels throughout each subject's whole brain. Specifically, to quantify shared-layout information, the same location between-city PS was calculated by correlating the remembered trials (in both spatial and temporal tasks) that share the same location from different cities using Pearson correlations. For example, the triads "Store 1-Store 2-Store 6" in City 1 were correlated with the triads "Store 1-Store 8-Store 9" in City 3 (Fig. 1a). In contrast, the different locations between-city PS was calculated by correlating the remembered trials (in both spatial and temporal task) that come from different locations in different cities. For example, the triads "Store 1-Store 2-Store 6" in City 1 were correlated with the triads "Store 1-Store 8-Store 5" in City 3 (Fig. 1a). Note, given that different location pairs usually contained perceptual differences (different stores), which in turn could contribute to lower PS for different locations than same locations, we matched the number of different stores between pairs when calculating PS. For example, there are two different stores between the triads "Store 3-Store 7-Store 6" and "Store 3-Store 7-Store 9" when calculating the same location between-city PS. Accordingly, when calculating the different locations between-city PS, we only consider the pairs that also involve two different stores, for example, the triads between "Store 1-Store 7-Store 6" and "Store 1-Store 7-Store 8". We transformed these similarity scores into Fisher's z-scores and compared the differences between the same location and different location pairs. Notably, we only included correctly retrieved trials into consideration and excluded any trials with any censored frames during the duration of the modeled GLM response using a framewise displacement threshold of 0.5 mm. A random-effects model was used for group analyses within the mPFC mask using a cluster-forming threshold of $Z > 2.6$, with $p < 0.05$ (corrected for family-wise error rate, using random field theory).

**Correlating frontal activity with hippocampal PS.** We also examined the role of prefrontal activity in modulating hippocampal PS during retrieval. Because the CA2/3/DG and CA1 were the regions that showed significant city-specific PS and conformed to our holistic hypothesis (see Results), we focused on these two regions and tested whether the ROIs could be modulated by PFC activity. We correlated the activation of each condition (unique city store within-city PS/unique city store between-city PS) in each voxel of the whole brain during retrieval with the corresponding PS in CA2/3/DG and CA1, separately. Because frontal activity was associated with the activity level in other brain regions, which was in turn associated with PS, we conducted a partial correlation analysis by correlating the activation level in each voxel across the whole brain and the corresponding PS of CA2/3/DG and CA1 while controlling for the activation levels of CA2/3/DG and CA1. The resulting Spearman's rank correlation coefficients were transformed into Fisher's z-scores and then directly compared between the unique store within-city vs. unique store between-city trials, which was put into further group analyses using a cluster-forming threshold of $Z > 3.1$, with $p < 0.05$ (corrected for family-wise error rate using random field theory, Fig. 2c).

**Classification analysis**
*Searchlight-based leave-one-city-out SVR classification.* To test whether the medial PFC preserved the shared spatial layout schema, we performed a linear Support Vector Regression (SVR)[108] using LIBSVM 3.12 (https://www.csie.ntu.edu.tw/~cjlin/libsvm/) as implemented in MATLAB (The MathWorks) to classify spatial distance using a searchlight approach[60]. Briefly, for each voxel, t-maps were extracted from a cubic ROI containing 343 surrounding voxels throughout each subject's whole brain. The idea of the leave-one-city-out classification was that if the mPFC could support shared layout schema, the spatial distances learned from

two cities should be able to generalize to the third new city. This is because the new city had the same layout as the two learned cities, even though the new city had new stores which they were not presented in the two learned cities (Fig. 1c). First, for each triad, we measured the physical distances by calculating the Euclidean distance between the top store displayed and each of the bottom stores (Fig. 1e). Then, we calculated the sum of the two Euclidean distances as the behavioral index of spatial distance of each triad. Notably, we only took correctly retrieved trials into consideration and excluded any trials with any censored frames during the duration of the modeled GLM response using a framewise displacement threshold of 0.5 mm. There were 18 possible Euclidean distances in all, and the Kolmogorov-Smirnov Test (KS-test) for group-level uniform distributions indicated that the frequency of the distance was not uniform ($t = 0.168$, $p = 0.016$, Supplementary Figure 5a) because there were too many trials in the shortest distance category (i.e., distance = 66) compared to the other distances. However, when the number of the shortest distance trials decreased from 14 to 11, the distribution became uniform ($t = 0.137$, $p = 0.08$, Supplementary Fig. 5b). Then, this step was performed for each individual participant to identify the proper number of the shortest distance trials to ensure a uniform distribution of distances ($Ps > 0.05$ in KS-Test). In each iteration of the leave-one-city-out cross-validation, a SVR model was trained on runs from two cities, which generated a prediction value of the runs of the third city based on each cubic ROI's activation patterns. The accuracy of the SVR prediction was then calculated as Spearman's rank correlation coefficient between actual and predicted values of the spatial distance index. The resulting correlation coefficients were transformed into Fisher's z-scores and then input into further group analyses within the mPFC mask using a cluster-forming threshold of $Z > 3.1$, with $p < 0.05$ (corrected for family-wise error rate, using random field theory). Similarly, the same SVR classification analysis was performed on hippocampal ROIs to test whether hippocampus could support shared spatial layout schema during spatial distance retrieval.

For temporal interval task, we also performed the searchlight-based leave-one-city-out SVM classification analysis across the whole brain. Similar to spatial distance task, for each triad, the sum of time durations between the top store (the center of the city) and each of the two bottom stores (i.e., $16 + 16 = 32$ s; $8 + 8 = 16$ s; $8 + 16 = 24$ s) was taken as the temporal interval index of the given triad (Fig. 1e). Since temporal interval could be divided into only three categories, here, a multi-class SVM classification was more appropriate to be adopted to decode the temporal interval information based on the activation patterns of each cubic ROI. In each iteration, we classify three categories (32 s/16 s/24 s) on two cities and tested on the left-out city. Classification accuracy thus represented the percentage of trials that were correctly categorized by the classifier. We balanced the number of trials in each condition of our classification analysis by randomly selecting the same number of trials for each condition (this procedure was performed both for training and testing set). The resulting classification accuracy map for all participants were input into further group analysis using a cluster-forming threshold of $Z > 3.1$, with $p < 0.05$ (corrected for family-wise error rate, using random field theory). Similarly, the same SVM classification analysis was performed on hippocampal ROIs to test whether hippocampus could support shared layout schema during time duration retrieval.

*ROI-based classification analysis for city.* To examine whether hippocampal subfields contained city-specific information, we performed an ROI-based multivoxel pattern classification analysis to classify three cities using a linear support vector machine (SVM)[109] using LIBSVM 3.12 (https://www.csie.ntu.edu.tw/~cjlin/libsvm/) implemented in MATLAB (The MathWorks). The classification analysis was conducted on 12 run-based t-maps and with a penalty parameter of 1. Since there were 4 runs per city, in each iteration of the leave-three-run-out cross-validation, we trained the classifier on 9 of retrieval runs (3 runs per city) and used the left out three runs (1 run per city) to test classification accuracy based on each hippocampal ROI's activation patterns. Specifically, for each iteration in the testing run, the SVM classifier generated a scalar probability estimate of the trial corresponding to 3 categories (City 1, City 2, and City 3). The category with the higher probability was then set as the classifier's prediction. Classification accuracy thus represented the percentage of runs that were correctly categorized by the classifier. We performed a group analysis on each hippocampus subfield using two-tailed, one-sample t-tests to determine whether the accuracy was above chance levels (i.e., 33.33%).

**Reporting summary**. Further information on research design is available in the Nature Research Reporting Summary linked to this article.

## Data availability
The authors declare that all behavioral, MRI data supporting the findings of this study are available at: https://osf.io/8qcr6/. The source data underlying all Figures and Supplementary Figures are provided as a source data file with this paper. Source data are provided with this paper.

## Code availability
Code to analyze the data is available at https://osf.io/8qcr6/.

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

## Acknowledgements

Funding for this project from NINDS/NIH grant NS076856.

## Author contributions

L.Z., E.A.I. and A.D.E. designed the experiment; L.Z. collected and analyzed the data. Z.Y.G. provided assistance with data analysis and interpretation. A.M. provided assistance with programming using Unity. L.Z. and A.D.E. drafted the manuscript and all authors edited the manuscript.

## Competing interests

The authors declare no competing interests.
