## [Peer Review File · Nature Communications]

Partially overlapping spatial environments trigger reinstatement in hippocampus and schema representations in prefrontal cortexREVIEWER COMMENTS

Reviewer #1 (Remarks to the Author):

Overall, this is a strong manuscript that uses fMRI in an attempt to take on several challenging questions about how memory functions and what the roles of the hippocampus and prefrontal cortex are. The manuscript is clear and the experimental design is clever. There are, however, several issues that detract from its impact in its current form. In particular:

- A fundamental premise in the paper is that the neural representation of these places can be reflected at the voxel level such that we can use the similarity structure across voxels to infer the similarity structure across neurons. We know, however, that even adjacent neurons in the hippocampus don't have similar place fields and each fMRI voxel will have tens of thousands of neurons in it. Even accepting the data at face value, the ability to infer from the fMRI data that the similarity structure here must reflect the similarity structure at the level of patterns across neurons is not clear and should at least come with a large caveat. As is, it reads as if there must be a one-to-one mapping here. Take the sentence (p 9), "In other words, when participants retrieve a neural representation for an environment, did they retrieve neural codes relevant to other stores in the environment as a whole in addition to those involved in the particular retrieval question?" At the very least, it is strongly implying they can directly measure the activity of neurons at these two time points and compare the patterns across events. Yet, they are working at a more abstract layer of representation than population or rate codes.

- In a related note, the "reversed" or "repulsion" representation findings all appear to come from fMRI studies. If we are to assume that these "reversed" patterns reflect a neural vector that points, not orthogonal to others, but in the opposite direction, it would be good to know if this has been found in recording studies. For example, this would amount to a place field "hole" in which activity at baseline exists but is suppressed upon entering a region. Of course, a localist representation of a place field is something of a strawman and this amounts to a reverse direction of a vector representation, but the logic is similar. Has any such representation been observed more directly? What exactly would this mean? The implication is one of inhibiting another representation, but it's not clear at all that we would be able to measure this form of active inhibition or repulsion using multivariate fMRI analysis.

- Here also, the authors are relying on observing an effect in CA1 but not in CA2/3/DG to argue for a difference between regions. Yet, the direct test for that difference is absent and, looking at the data, unlikely to be reliable. For example, one point (the high point) in the Fig 4d plot in CA1 Unique (or the low point in CA2/3/DG Unique) appears to be able to match the patterns seen in CA1 vs. CA2/3/DG. This makes claims in the discussion such as "At the same time, we found evidence for a different mechanism within CA1" difficult to support so strongly.

- The authors have data from the entire brain. How unique are the subfield-level findings here? Do these same patterns exist outside of the MTL? While I appreciate the a priori nature of the analyses here, the authors are claiming key roles for hippocampal subfields in allowing us to perform specific memory functions. To establish this, logically, we would need to know what the prior probability of these patterns are in the fMRI data. As the authors have specific patterns they are looking for, it would seem relatively straightforward to assess how well activity in any region correlates with that pattern, sweep this across the brain, and derive a null distribution that the current findings could be compared against.

Reviewer #2 (Remarks to the Author):

In their manuscript, Zheng et al. present the results of a spatial learning task designed to target encoding mechanisms of the hippocampus and prefrontal cortex. By presenting participants with highly structured cities consisting of spatially organized stores, they manipulated overlap between

learning experiences in terms of space, time, and category (i.e., store type). Interestingly, they find environment-specific coding in hippocampal fMRI patterns with evidence for differentiated representations in spite of the partial overlap between environments. They also find that activity in lateral PFC tracks hippocampal pattern similarity for trials with environment-unique reinstatement and that mPFC codes for spatial layout across cities, suggesting a role for building more schema-like representations.

Characterizing the factors of hippocampal encoding, especially those factors that define episodic experience, is a hot topic. Understanding specifically how the hippocampus and its related cortical networks both captures unique experiences and generalizes across related information in those experiences is a key question in the field. In a crowded area, this work stands out for its well developed paradigm and empirical approach. Indeed the spatial navigation task offers both the carefully controlled experimental manipulations necessary to pinpoint neurocognitive mechanisms and a bridge towards more naturalistic everyday experiences. The introduction and discussion are well written, although the results are much rougher. Overall, I find the results highly interesting, but several issues with general presentation, the rationale for the different analyses, and missing methodological details make me hesitate:

1) In contrast to the well-crafted paradigm, the overall analytic approach does not have a clear logical thread and seemingly lacks proper rationale. Rather, the results are presented as a collection of loosely-related findings each employing different analytic approaches in different brain regions. For example, the analyses begin with a searchlight classification throughout MTL which then switches to ROI-based subfield MPS analyses. Presumably the tSNR voxel selection in the MPS constrains the MPS analysis to similar subsets of voxels as those identified in the initial classification analysis, but there remains a large disconnect.

After switching the searchlight to subfield-based analyses, there is another switch to whole brain voxelwise correlations before coming back to ROI-based analyses for the schematic spatial layout SVR. Why were these analytic choices made? I can imagine that there are good reasons (e.g., some are more theoretically driven like the subfield MPS, others more exploratory like the correlation analysis), but this should be clarified throughout the presentation of the results (and methods).

Another example, quite unexpectedly, subiculum is included in the SVR analysis of schematic spatial layout. Whereas a reasonable argument could be (and was) made through out justifying the ROI-based analyses that included CA1 and CA2/3/DG subfields due to the classification results in Figure 2, the inclusion of subiculum in this final analysis comes with no explanation. Given the focused nature of the preceding analyses, including subiculum only in the final analysis should come with a clear justification.

On one hand, the authors have truly flexed their analytic skills in leveraging all of these methods. However, the presentation of the work reads more like a list of analyses that worked rather than a clear set of analytic steps that form a logical chain. I want to be clear here that I think this issue is mostly due to presentation and that a careful revision that focus on the motivation of each analysis could greatly strengthen the manuscript.

2) That activation in RIFG correlated with Unique store within-city PS is certainly interesting, especially given the role of lateral PFC in conflict resolution and mediating difficult memory retrieval. But, it seems to be a relatively weak effect (27 voxels). I must be missing something, but I am unsure exactly how this cluster (and the other clusters noted in the supplemental materials) were found. Was a cluster extent threshold employed? If so, $P=0.01$ ($Z=2.3$) is quite a liberal threshold. To fully understand and evaluate this effect, the specific methodology needs to be clarified. Also, the discussion treats this finding with quite strong language. For example, on lines 645-648 the authors write that "...interactions with lateral PFC are critical for accentuating the fidelity of the reinstated stores...". Of course, whether or not lateral PFC is critical to this process cannot be determined with the sort of correlative methods in the current work.

3) The SVR analysis identified mPFC as coding schema representations of layout is quite novel. I am intrigued by the degrees of freedom in the summed Euclidean distance measure used to characterize the trials—how many different values were possible, how many trials shared the same index value, and more generally, how much does this quantification reveal about the spatial layout? Although the findings are interesting, I am curious why this SVR approach was used rather than the more typical MPS analysis that could potentially be used to derive a more detailed quantification of the spatial layout from multidimensional scaling of similarity relationships in neural patterns. For example, the distance index would not discriminate store triads with mirrored spatial arrangements, but may be uncovered in pattern similarity.

4) I dislike criticisms based simply on sample size. But, I feel that it is a true limitation of the current work. Only having 18 participants in the final sample limits the power of the analysis. And, this is reflected in the generally weaker effects reported throughout the manuscript. This is unfortunate because I think all the ingredients are there for a compelling demonstration of hippocampal reinstatement, repulsion with overlapping information, and mPFC schematic codes. But, the data is limited due to the smaller sample size.

Minor comments

- The description of the MPS hypotheses in the Results section are difficult to parse. Rewriting these to be more concise would help with the flow the manuscript.
- A breakdown of the total number of trials going into each condition of the MPS analysis would be helpful.
- Line 904: "portions" instead of "potions"
- Line 905: missing space between the and 10th, but maybe there's additional text in this sentence that should be removed?
- Making the ROI shading transparent in Figure 2 would increase the visibility of the highlighted clusters.

Reviewer #3 (Remarks to the Author):

The authors present a really interesting study of how shared features between different environments influence representations in the hippocampus and related cortical areas (particularly PFC). Some significant strengths include the clever extension of their prior work/design in this area, and controlling for factors such as tSNR differences between ROIs in the MPS analyses. The differentiation in results for stores unique to a city vs those recurring in multiple cities is exciting and controls for various factors such as similarity being driven by temporal experience proximity at encoding and not spatial context per se. The tests are theory driven, and the correspondence of the results to this strong framework position them to advance the field in important ways. Overall, I greatly enjoyed reading this manuscript.

I did, however, have a number of concerns - some more minor than others - which I am optimistic can be addressed fully.

I found figure 1 a-c very confusing – as far as I can tell, 1a is the ground truth for the configural structure, and 1b is a venn diagram and – in contrast to a+c – not a map. There is no physical overlap between environments, but some content / landmark overlap and configural overlap for repeated instances of these landmarks. But 1b, at first blush, could imply otherwise and even suggest something more akin to an A-B B-C A-C associative learning problem, where there is continuity and one might learn the spatial relationship between 4 (in City 1) and 9 (in City 3) via their shared landmark buildings in the layout (as depicted) in the regions of overlap. But that is a very different task, if I understand correctly. I think the authors could improve presentation substantially for naive readers through some very small changes to that opening task figure.

I had some concerns about how statistics were reported and in some cases computed. It may be a

matter of simple clarification in many cases.

--Firstly, I think it is important for the authors to report coordinates from their hippocampal searchlight clusters, which will allow readers to relate these outcomes to prior work and anatomical boundaries.

--I was confused and concerned about how significance testing was reported - the Methods only indicate a cluster-based significance threshold of $p < 0.01$ based on permutation testing - but at what voxel-wise threshold? Was this $p < 0.05$ relative to 33% classification, for example, for the first hippocampal result reported, being a 3-env classifier? There has been some very lively discussion in the field in recent years, and strong skepticism, for applying cluster-based correction of this sort when using such a liberal height p/t threshold in unsmoothed fMRI data, where assumptions of cluster correction break down. A cluster of just 25 voxels, identified hi-res whole-brain at $p < 0.05$ classification significance, would fall deep into firing zone of those arguments, I think. The authors should clarify, and if this is indeed the case should say more about the reliability of these results in light of that concern.

--Similarly, I found the references to Bonferroni correction in the same early results on hippocampal coding (lines 238+) unclear. Did the authors use the peak value from the cluster identified in the searchlight? Or perhaps the average? What was the Bonferroni correction applied against (from the t -values it is clearly not the whole-brain voxel count, but I was otherwise unsure)? The answers to these questions would also help me appreciate the extent to which this analysis avoids circularity

--Why were such tests conducted one-tailed, given that classification can be below chance?

--Similar concerns applied to the RIFG and Frontal Pole in terms of significance and circularity in follow-up tests (e.g., line 380+)

--My final statistical concern related to the idea of "representational repulsion". On the one hand, it was interesting and encouraging to see some replication and extension of this idea from the recent work from Chanales et al. On the other hand, it is not clear to me that "repulsion" is mechanistically distinct from orthogonal - particularly at the granular level of fMRI, where it is not possible to measure the firing patterns of specific neurons. The authors make a strong distinction there in the manuscript, but my concern rests on two considerations: 1) raw similarity scores are notoriously influenced by tSNR and univariate signal among other things, and so interpretation of a raw magnitude, rather than relative, metric of representation is suspect. 2) although the change in sign seems less ambiguous in meaning, I'm not sure it is, at least at this level of measurement. In traditional RSA work, r is taken as a continuous index of distance from one representation to another and a negative correlation is simply greater distance in a distance matrix and this broad approach can have notable correspondence to neural data (e.g., Kriegeskorte ... Bandettini, 2008, Neuron). If the authors feel strongly that an important change happens from neural patterns approaching orthogonal to passing it (an idea I do find interesting), it feels some theory and methods work may be needed to flesh that out and bolster the argument that it should be interpreted that way.

I found the frontal pole results quite novel, as this area has received little mechanistic attention in navigation work. Given evidence here for this region's relationship to enhancing within-env fidelity and minimizing cross-env interference, I was surprised the authors didn't cite some of the sparse work tying this brain area to replanning+strategization surrounding specific locations/landmarks at retrieval (as reviewed by Spiers and Gilbert, 2015 Frontiers; see also a new review to just come out: Patai ... Spiers 2021 TiCS), and specific examples linking the area to landmark/goal representation in the hippocampus and to route planning (Brown ... Wagner, 2016 + 2020 Science and Curr Bio)

I don't think the authors should summarize and refer to their FPC results as dorsolateral PFC - these areas are different in cytoarchitecture, connectivity, and functional associations in prominent theories of PFC organization (e.g., Badre and Nee, 2018, TiCS; Nee and D'Esposito, 2016, eLife) and it may

confuse the literature to do so – depending on where the coordinates fall at least (although this raised for me some uncertainty on where the clusters really fall? Could the authors clarify in revision?)

In contrast to the IFG+FPC outcomes which were identified voxel-wise, the authors report outcomes from a very large region of mPFC. One concern is that the underlying functional anatomy is highly heterogeneous (e.g., subdivisions of this mPFC ROI have been tied to goal distance signals and goal-specific coding in hippocampus by various groups; other parts of this ROI, as depicted, encompass anterior cingulate cortex; parts with strong hippocampal connectivity and parts with virtual no direct connectivity with the hippocampus; etc). Another concern is that with more voxels, the ability to detect correlations may be more robust, and it is somewhat challenging to contrast this outcome with those in other prefrontal areas. Although I would advocate a more precise ROI, I would at least encourage a searchlight analysis to help understand what anatomical loci underlie the overall outcome from this very large region.

We thank the reviewers for their helpful suggestions and consideration of our manuscript. Critically, in response to concerns from reviewers, we have recruited an additional 12 participants, with 9 of them added into the final analysis, thereby significantly increasing our sample size (new sample size = 27 participants). Importantly, all of our effects remain robust, and, in many cases, are now stronger. In addition, we have redone all of our original analyses, redone all the figures, and added several new analyses on the new dataset according to the reviewers' suggestions. We believe that our manuscript is now significantly improved as a result of their insightful suggestions. We highlight the major modifications and additional analyses we have conducted below, followed by specific responses to individual reviewers.

Summary of major modifications and additional analyses in the revised manuscript

1. We rearranged the results in response to reviewer concerns about the logical flow of the manuscript: we put the section titled: "Holistic representation in the hippocampus" first and moved the section titled: "Spatial memory representations in the hippocampus are environment-specific" to the end after discussion of schemas.
2. We added a two-way repeated measures ANOVA to compare the difference in PS between CA1 and CA2/3/DG in the repulsion vs. pattern separation analysis (Results) in response to reviewer concerns about subfield differences.
3. We employed a new measure, the relative difference value of between-city PS for three shared city trials vs unique city trials, in addition to raw similarity scores, to better represent the "repulsion effect" (Fig.3e) in response to reviewer concerns about the robustness of our findings.
4. We added a searchlight-based leave-one-city-out cross-validation SVR classification analysis across the whole brain (Fig.4a, b, c, and d) in response to reviewer concerns about the generality of our findings outside of our originally selected ROIs.
5. We added a searchlight-based MPS analysis to detect schematic spatial layout information across the whole brain, with a particular focus on the mPFC (Fig.4e), in response to reviewer concerns about the specificity of our schema-related findings.
6. We added an ROI-based MPS analysis on hippocampal subfields to detect schematic spatial layout information (Supplementary Fig.6c) in response to reviewer concerns about our schema analysis.
7. We added an ROI-based SVM classification analysis, replacing our searchlight-based SVM classification analysis, to detect environment-specific representations (Fig.5b) in response to reviewer concerns about the logical thread of our analytic methods.
8. We added the number of trial pairs that went into each condition for each MPS analysis (Supplementary Fig.2c and Supplementary Fig.4c).

9. We added a uniformity test for the distribution of Euclidean spatial distances (Supplementary Fig.5a and b)

Response to Review #1

Overall, this is a strong manuscript that uses fMRI in an attempt to take on several challenging questions about how memory functions and what the roles of the hippocampus and prefrontal cortex are. The manuscript is clear and the experimental design is clever. There are, however, several issues that detract from its impact in its current form. In particular:

RI.1)- A fundamental premise in the paper is that the neural representation of these places can be reflected at the voxel level such that we can use the similarity structure across voxels to infer the similarity structure across neurons. We know, however, that even adjacent neurons in the hippocampus don't have similar place fields and each fMRI voxel will have tens of thousands of neurons in it. Even accepting the data at face value, the ability to infer from the fMRI data that the similarity structure here must reflect the similarity structure at the level of patterns across neurons is not clear and should at least come with a large caveat. As is, it reads as if there must be a one-to-one mapping here. Take the sentence (p 9), "In other words, when participants retrieve a neural representation for an environment, did they retrieve neural codes relevant to other stores in the environment as a whole in addition to those involved in the particular retrieval question?" At the very least, it is strongly implying they can directly measure the activity of neurons at these two time points and compare the patterns across events. Yet, they are working at a more abstract layer of representation that population or rate codes.

We appreciate the reviewer pointing out this issue; we certainly did not intend to express or imply any connection between BOLD and single neuron activity. In fact, the lack of a direct mapping between hippocampal BOLD and single neuron activity has been a topic of past publications in the lab and is certainly an issue we are sensitive to¹⁻³. While some studies have suggested similar phenomenon at the level of BOLD and population-level neural activity^{4,5}, these are likely different phenomenon of unknown linkage⁶. As we have argued recently, however, multivariate pattern analysis (MVPA) employs the distributed patterns of voxels to make inferences about information content, and therefore may represent one possible means for better linking BOLD signal changes to the *information* coded by underlying neural activity⁷, with some studies suggesting clearer linkages between MVPA and population-level decoding of single neuron activity⁸. To try to deal with this issue, we edited the sentence mentioned by the reviewer and also added the following to the discussion:

"In other words, when participants retrieve a neural representation for an environment, do such codes also contain information relevant to other stores in the environment as a whole in addition to those involved in the particular retrieval question?"

Added to the discussion:

“Although our study did not involve single neuron recordings, and therefore may have involved different forms of neural-related information coding schemes, we did not find evidence for spatial schema within the hippocampus.”

R1.2)- In a related note, the “reversed” or “repulsion” representation findings all appear to come from fMRI studies. If we are to assume that these “reversed” patterns reflect a neural vector that points, not orthogonal to others, but in the opposite direction, it would be good to know if this has been found in recording studies. For example, this would amount to a place field “hole” in which activity at baseline exists but is suppressed upon entering a region. Of course, a localist representation of a place field is something of a strawman and this amounts to a reverse direction of a vector representation, but the logic is similar. Has any such representation been observed more directly? What exactly would this mean? The implication is one of inhibiting another representation, but it’s not clear at all that we would be able to measure this form of active inhibition or repulsion using multivariate fMRI analysis.

We appreciate the reviewer bringing this point to our attention. To the best of our knowledge, there are no recording studies in rodents that have found analogous “repulsion” representations. When rodents are put in two different environments, the correlation between the firing activity of place cells in the two environments is near zero, termed “remapping,” and is likely related to pattern separation⁹. Previous rodent studies found that even when two environments were highly similar, place cells fired in distinct ways, but not in a negative relationship¹⁰. Although inhibiting another representation is a plausible mechanism underlying “reversed” representation, direct evidence from recording studies is lacking.

Our “reversed” patterns found here are instead consistent with several human fMRI studies^{11,12} and also consistent with theoretical models suggesting that CA1 may represent changes in input in a linear fashion¹³, by which its representational space/range is large enough to allow “reverse” representations to occur—even past the point of orthogonalization. To address this issue, we have added the following to the discussion:

“Such reverse representations have been reported previously for overlapping routes during virtual navigation using fMRI¹¹ although, to the best of our knowledge, not in any previous rodent single neuron studies.”

R1.3)- Here also, the authors are relying on observing an effect in CA1 but not in CA2/3/DG to argue for a difference between regions. Yet, the direct test for that difference is absent and, looking at the data, unlikely to be reliable. For example, one point (the high point) in the Fig 4d plot in CA1 Unique (or the low point in CA2/3/DG Unique) appears to be able to match the patterns seen in CA1 vs. CA2/3/DG. This makes claims in the discussion such as “At the same time, we found evidence for a different mechanism within CA1” difficult to support so strongly.

We thank the reviewer for this suggestion. We agree that a direct comparison between CA1 and CA2/3/DG is necessary to draw a conclusion related to different mechanisms within CA1. Therefore, we performed a two-way repeated measures ANOVA, with the factor of ROI and Condition as within-subjects variables. We found a significant condition by ROI interaction effect, suggesting that the overlapping stores for unique and shared cities involved different mechanisms. Notably, we did not find a main effect of ROI, suggesting that differences in signal within the subfields was not driving the effect. To address this issue, we have added the following to the Results section:

“The different pattern between CA1 (i.e., repulsion) and CA2/3/DG (i.e., pattern separation) was confirmed by a significant ROI (i.e., CA1, CA2/3/DG) by condition (between-city PS for unique city trials, between-city PS for three shared city trials) interaction effect ($F(1,26) = 4.416$, $p = 0.045$). Importantly, there was no main effect between the two ROIs ($F(1,26) = 0.545$, $p = 0.467$), and the observed repulsion effect in CA1 (but not in CA2/3/DG) could not be accounted by higher between-city PS for unique city trials in CA1 because there was no significant difference for between-city PS for unique city trials between CA1 and CA2/3/DG ($t(26) = 1.800$, $p = 0.117$, two-tailed). ”

We have also tried to soften our statement somewhat:

“Together, these results indicate that both CA2/3/DG and CA1 contributed to the discrimination of overlapping memories but through potentially different mechanisms.”

4- The authors have data from the entire brain. How unique are the subfield-level findings here? Do these same patterns exist outside of the MTL? While I appreciate the a priori nature of the analyses here, the authors are claiming key roles for hippocampal subfields in allowing us to perform specific memory functions. To establish this, logically, we would need to know what the prior probability of these patterns are in the fMRI data. As the authors have specific patterns they are looking for, it would seem relatively straightforward to assess how well activity in any region correlates with that pattern, sweep this across the brain, and derive a null distribution that the current findings could be compared against.

We agree that it is important to examine whether the findings we found exist outside the MTL. To address this concern, first, we tested whether the representation of other brain regions outside the MTL are also holistic by using both ROI-based and searchlight-based MPS. In the ROI-based MPS analysis, we created ROIs using a meta-analysis in Neurosynth (<https://neurosynth.org/>) with the key term “memory retrieval”. Based on the Neurosynth associative tests, a threshold Z-score > 3.1 was applied to create clusters. There were 9 ROIs, including middle frontal gyrus, inferior frontal gyrus, medial frontal cortex, frontal pole, lateral occipital cortex, precuneus, paracingulate, middle temporal gyrus, and temporal pole. However, the results indicated that no ROIs showed significantly higher within-city PS for unique stores than between-city PS for unique stores ($P_s > 0.121$). Similarly, no cluster was found in a searchlight-based MPS across the whole brain. Therefore, all further analyses were focused on hippocampal subfields.

Overall, our findings related to holistic representation and repulsion effects appear specific to the hippocampus, at least within our data.

We have added the following to the results to clarify this issue:

“Finally, we also examined whether holistic representations are also present outside the MTL by using both ROI-based and searchlight-based MPS analyses (Supplementary Note3). No ROIs or clusters showed holistic representation-related effects outside the MTL. Therefore, all further analyses were focused on hippocampal subfields.”

Response to Reviewer #2:

In their manuscript, Zheng et al. present the results of a spatial learning task designed to target encoding mechanisms of the hippocampus and prefrontal cortex. By presenting participants with highly structured cities consisting of spatially organized stores, they manipulated overlap between learning experiences in terms of space, time, and category (i.e., store type). Interestingly, they find environment-specific coding in hippocampal fMRI patterns with evidence for differentiated representations in spite of the partial overlap between environments. They also find that activity in lateral PFC tracks hippocampal pattern similarity for trials with environment-unique reinstatement and that mPFC codes for spatial layout across cities, suggesting a role for building more schema-like representations.

Characterizing the factors of hippocampal encoding, especially those factors that define episodic experience, is a hot topic. Understanding specifically how the hippocampus and its related cortical networks both captures unique experiences and generalizes across related information in those experiences is a key question in the field. In a crowded area, this work stands out for its well-developed paradigm and empirical approach. Indeed, the spatial navigation task offers both the carefully controlled experimental manipulations necessary to pinpoint neurocognitive mechanisms and a bridge towards more naturalistic everyday experiences. The introduction and discussion are well written, although the results are much rougher. Overall, I find the results highly interesting, but several issues with general presentation, the rationale for the different analyses, and missing methodological details make me hesitate:

R2.1) In contrast to the well-crafted paradigm, the overall analytic approach does not have a clear logical thread and seemingly lacks proper rationale. Rather, the results are presented as a collection of loosely-related findings each employing different analytic approaches in different brain regions. For example, the analyses begin with a searchlight classification throughout MTL which then switches to ROI-based subfield MPS analyses. Presumably the tSNR voxel selection in the MPS constrains the MPS analysis to similar subsets of voxels as those identified in the initial classification analysis, but there remains a large disconnect.

After switching the searchlight to subfield-based analyses, there is another switch to whole brain voxelwise correlations before coming back to ROI-based analyses for the schematic spatial

layout SVR. Why were these analytic choices made? I can imagine that there are good reasons (e.g., some are more theoretically driven like the subfield MPS, others more exploratory like the correlation analysis), but this should be clarified throughout the presentation of the results (and methods).

Another example, quite unexpectedly, subiculum is included in the SVR analysis of schematic spatial layout. Whereas a reasonable argument could be (and was) made through out justifying the ROI-based analyses that included CA1 and CA2/3/DG subfields due to the classification results in Figure 2, the inclusion of subiculum in this final analysis comes with no explanation. Given the focused nature of the preceding analyses, including subiculum only in the final analysis should come with a clear justification.

On one hand, the authors have truly flexed their analytic skills in leveraging all of these methods. However, the presentation of the work reads more like a list of analyses that worked rather than a clear set of analytic steps that form a logical chain. I want to be clear here that I think this issue is mostly due to presentation and that a careful revision that focus on the motivation of each analysis could greatly strengthen the manuscript.

We appreciate the reviewer's suggestions of a clearer logical thread for both the results and analytic methods. We have rewritten the results in several different sections to try to accomplish this goal. We've also performed some new analysis to make our approach more consistent throughout the manuscript. We have also rearranged the ordering of the results so the logic for each choice of analysis is clearer.

1, Within the MTL, all the analyses, including MPS and classification analyses, were performed using an ROI-based approach. This allowed us to apply the t-SNR methods to exclude potential influences from voxels with poor or highly variable signal.

2, Outside the MTL, with our results in analyses focused on the mPFC, we performed all the MPS and classification analyses using a whole-brain searchlight approach. Given that the mPFC is highly functionally heterogeneous, we believe that it is more suitable to apply searchlight methods to identify precise clusters in mPFC.

3, As an exploratory analysis, we correlated the PS of each hippocampal subfield and the activation of each voxel of the whole brain. This method allowed us to detect, in an unbiased manner, which voxels outside of the hippocampus might be modulating hippocampal environment-specific signals.

Based on the analysis principles highlighted above, in the revision, we rearranged the results in a new way to better highlight the novelty of this manuscript and at the same time, provide a clearer thread for the analysis choices. As the most central point of our manuscript, we moved the section "Holistic representation in the hippocampus" to the first part of the results. Here, we included all 6 ROIs within the MTL in the MPS analysis to test whether the representations of the MTL were holistic. After finding that CA1 and

CA2/3/DG showed evidence for holistic retrieval, we then focused on these two subfields (CA1 and CA2/3/DG) to test the repulsion hypothesis. Next, based on the whole brain searchlight, we tested whether mPFC or hippocampus played a role in schema representation. We agree with the reviewer that there is no need to include subiculum into consideration since this subfield was not significant in any of the preceding analyses. However, whether a region supports spatial schema has no direct relationship with whether a region supports environment-specific or holistic representation. Hence, the subiculum was included in the SVR analysis to avoid missing meaningful findings in this region.

We hope that the modified analyses above and significant reorganization of the manuscript helps deal with the reviewer concerns about logical flow.

R2.2) That activation in RIFG correlated with Unique store within-city PS is certainly interesting, especially given the role of lateral PFC in conflict resolution and mediating difficult memory retrieval. But, it seems to be a relatively weak effect (27 voxels). I must be missing something, but I am unsure exactly how this cluster (and the other clusters noted in the supplemental materials) were found. Was a cluster extent threshold employed? If so, $P=0.01$ ($Z=2.3$) is quite a liberal threshold. To fully understand and evaluate this effect, the specific methodology needs to be clarified. Also, the discussion treats this finding with quite strong language. For example, on lines 645-648 the authors write that "...interactions with lateral PFC are critical for accentuating the fidelity of the reinstated stores...". Of course, whether or not lateral PFC is critical to this process cannot be determined with the sort of correlative methods in the current work.

We apologize for not making our analysis clearer in the original submission. In the revision, we correlated the activation of each condition (unique city store within-city PS / unique city store between-city PS) in each voxel of the whole brain during retrieval with the corresponding PS in CA2/3/DG while controlling for the activation level of hippocampal subfields. The partial correlation coefficients were transformed into Fisher's z-scores and we focused on the differences between unique store within-city PS vs. unique store between-city PS comparison. Then, we employed a random-effects model in the group analysis using a cluster-based threshold of $Z > 3.1$, with $p < 0.05$ (corrected using the family-wise error rate with random field theory, see Methods). We found a significant cluster located in LIFG (MNI: -52, 22, -2, $Z = 3.91$, number of voxels in cluster = 99, please see figure below).

Of note, in the new dataset, the partial correlation analysis revealed LIFG was significant, but not RIFG, which is what we found in the previous version of the manuscript. To be clear, we continue to find a cluster in RIFG overlapping with our previous one (please see the figure below; RIFG cluster [in red] overlapped with our previous result [in blue]). Considering the larger sample size employed in our revision (N=18 in the previous version and N=27 in the current version), and the fact that we did not have any *a priori* hypotheses related to laterality, we believe that this new result in LIFG is more robust and have chosen to report this in our revised manuscript.

LIFG, which belongs to lateral PFC, is often considered part of a larger set of regions involved in cognitive control¹⁴. We believe that one possible interpretation of the positive correlation between IFG activity and holistic retrieval in the hippocampus is that the lateral PFC facilitates spatial holistic retrieval by enhancing the fidelity of the same city's representation and minimizing those from other cities or those shared across cities. We agree, however, that the correlational methods we have employed cannot provide the directionality or information on causality, although this issue could be tested in future brain stimulation studies.

Therefore, we have toned down any implied causality by rephrasing this sentence, which now reads: "Our findings therefore suggest a new possible framework for spatial holistic retrieval: the hippocampus is not the only region responsible for incidental

reinstatement of unique information within an environment, and interactions with the lateral PFC could provide a mechanistic basis for accentuating the fidelity of the reinstated stores as part of mediating interference during successful retrieval.”

We have also added the following to the Discussion:

“Neurostimulation or lesion studies would be necessary, however, to determine the directionality of this potential influence. Another possibility, although less likely given the suggested role of PFC in cognitive control, is that hippocampal subfields modulate schema-like representations in the PFC or both regions interact in some to-be-defined manner during holistic retrieval.”

R2.3) The SVR analysis identified mPFC as coding schema representations of layout is quite novel. I am intrigued by the degrees of freedom in the summed Euclidean distance measure used to characterize the trials—how many different values were possible, how many trials shared the same index value, and more generally, how much does this quantification reveal about the spatial layout? Although the findings are interesting, I am curious why this SVR approach was used rather than the more typical MPS analysis that could potentially be used to derive a more detailed quantification of the spatial layout from multidimensional scaling of similarity relationships in neural patterns. For example, the distance index would not discriminate store triads with mirrored spatial arrangements but may be uncovered in pattern similarity.

We thank the reviewer for this constructive suggestion. There were 18 different Euclidean distances in this experiment and the distribution of the distances are shown in Supplementary Fig.5 (please also see below). The chances of each store being presented in the whole experiment were nearly the same to ensure the distances between stores could be spread equally across the environment. To ensure, however, that the distances were spread out the same, we performed an additional analysis by shortening the number of the shortest distance (i.e., distance = 66) such that the distribution of distances was now uniform. In the revision, we performed SVR classification on these distances and have added this information to the Methods and Supplementary Material.

Supplementary Fig. 5. The distributions of Euclidean distances. a The group level distribution of Euclidean distances across trials was not uniform ($p = 0.016$), with shorter distances overrepresented. b After randomly removing 3 trials of the shortest distance, the distribution was uniform ($p = 0.08$).

We agree with the reviewer that an MPS analysis should also be able to identify schematic spatial layout representations. Here, we performed the recommended analysis by running a searchlight-based MPS analysis. Specifically, if mPFC supports such schematic spatial layout information as we found in the SVR analysis, the between-city PS for the same location pairs should be higher than between-city PS for different location pairs (see Methods). As predicted, this MPS analysis revealed a significant cluster located in the medial frontal pole (MNI: -8,62, 28, $Z = 3.98$, Fig.4e, please see figure below), although only at a lower threshold ($Z = 2.6$) and using a small volume correction within the mPFC mask. We have added this analysis to the results in the revision:

“In addition to the SVR classification analysis, we also tested whether MPS might reveal quantitatively similar relationships in neural patterns for shared locations. Therefore, we tested the spatial schema hypothesis by performing a searchlight-based MPS throughout whole brain (see Methods), predicting that if the mPFC contained shared layout information across cities, the same location between-city PS would be higher than different location between-city PS. As predicted, this analysis revealed a significant cluster located in medial frontal pole (MNI: -8,62, 28, $Z = 3.98$, Fig.4e and Supplementary Fig.6d; note that we applied a lower threshold ($Z = 2.6$) and a small volume correction within mPFC mask).”

Fig.4e The searchlight MPS revealed a significant cluster located in medial frontal pole (MNI: -8, 62, 28, $Z = 3.98$) whose between-city PS for the same locations was higher than for different locations.

R2.4) I dislike criticisms based simply on sample size. But, I feel that it is a true limitation of the current work. Only having 18 participants in the final sample limits the power of the analysis. And, this is reflected in the generally weaker effects reported throughout the manuscript. This is unfortunate because I think all the ingredients are there for a compelling demonstration of hippocampal reinstatement, repulsion with overlapping information, and mPFC schematic codes. But, the data is limited due to the smaller sample size.

We thank the reviewer for this important point. We have added another 9 participants to the final analysis and redone all the of the analyses accordingly. Our results are more robust and stronger as a result of increasing our sample size and we thank the reviewer for this suggestion.

Minor comments

R2.5) The description of the MPS hypotheses in the Results section are difficult to parse. Rewriting these to be more concise would help with the flow the manuscript.

We agree with the reviewer that it is important to better phrase the MPS hypotheses. We have attempted to clarify and shorten this section:

“We sought to test whether neural codes within the hippocampus are holistic. In other words, when participants retrieve a neural representation for an environment, do such codes also contain information relevant to other stores in the environment as a whole in addition to those involved in the particular retrieval question? If the spatial retrieval is specific to a trial (i.e., non-holistic), we predict the correlation between that trial and “incidental” unrepresented stores specific to that city should be comparable to the correlation between that trial and unrepresented stores belonging to other cities (Fig.2a, left panel). In contrast, if spatial retrieval is holistic, we predict a higher correlation between neural patterns retrieved for a specific trial and unrepresented stores for that same city (i.e., unique/shared store within-city PS) compared to

those specific to other cities (i.e., unique store between-city PS; Fig.2a, middle panel). A corollary to this is whether the “incidental” stores retrieved (i.e., those not in the retrieval question) are unique to that environment (differentiated) and not the ones shared across environments (non-differentiated). We lay out the possibilities for shared stores (i.e., holistic and non-differentiated vs. holistic and differentiated) in Fig.2a.

R2.6) A breakdown of the total number of trials going into each condition of the MPS analysis would be helpful.

We've added this information in Supplementary Fig.2c and Supplementary Fig.4c. We note that the number of pairs going into each condition were not unequal, therefore, we conducted a control analysis to match the trial pairs of each condition by randomly resampling the pairs of the condition which had more pairs with the smaller number of pairs 5000 times. Our results do not change. Please see Supplementary Note 4, Note 5, Supplementary Fig.2, and Supplementary Fig.4.

R2.7) Line 904: "portions" instead of "potions"

We thank the reviewer for catching this error, which has now been corrected.

- Line 905: missing space between the and 10th, but maybe there's additional text in this sentence that should be removed?

We thank the reviewer for catching this error, which has now been corrected.

R2.8) Making the ROI shading transparent in Figure 2 would increase the visibility of the highlighted clusters.

Thank you for the suggestions. We have replaced this searchlight analysis with an ROI analysis.

Response to Reviewer #3

The authors present a really interesting study of how shared features between different environments influence representations in the hippocampus and related cortical areas (particularly PFC). Some significant strengths include the clever extension of their prior work/design in this area, and controlling for factors such as tSNR differences between ROIs in the MPS analyses. The differentiation in results for stores unique to a city vs those recurring in multiple cities is exciting and controls for various factors such as similarity being driven by temporal experience proximity at encoding and not spatial context per se. The tests are theory driven, and the correspondence of the results to this strong framework position them to advance the field in important ways. Overall, I greatly enjoyed reading this manuscript.

I did, however, have a number of concerns - some more minor than others - which I am optimistic can be addressed fully.

R3.1 "I found figure 1 a-c very confusing – as far as I can tell, 1a is the ground truth for the

configural structure, and 1b is a venn diagram and – in contrast to a+c – not a map. There is no physical overlap between environments, but some content / landmark overlap and configural overlap for repeated instances of these landmarks. But 1b, at first blush, could imply otherwise and even suggest something more akin to an A-B B-C A-C associative learning problem, where there is continuity and one might learn the spatial relationship between 4 (in City 1) and 9 (in City 3) via their shared landmark buildings in the layout (as depicted) in the regions of overlap. But that is a very different task, if I understand correctly. I think the authors could improve presentation substantially for naive readers through some very small changes to that opening task figure.”

We thank the reviewer for bringing up Fig.1b as a point of confusion. As the reviewer pointed out, there was no physical overlap between the three environments. We realized that the background color of the three environments in Fig.1b was the same as in Fig.1a, which may have given the erroneous impression that the three cities overlapped with each other physically. Here, we have changed the background color to transparent in Fig.1b, and rewritten the corresponding legend (please see below and also Fig.1b in manuscript). We hope that these changes will better convey the differences between physical and conceptual (abstract) overlap.

Schematic of the experimental design. **a** Layouts for each environment are depicted from a survey view. Each colored box represents a target store (except the center one). Cities 1 & 2 are identical aside from the “Store 4” in City 1, which was changed to “Store 7” in City 2. Cities 2 and 3 were the same except “Store 2” and “Store 6” in City 2 were changed to “Store 8” and “Store 9” in City 3, respectively. The lines from the center store to each peripheral store represent the temporal durations: 8 s (red solid line); 16 s (red dashed line). **b** Venn diagram depicting the shared stores between the three overlapping environments (note: this does not show physical overlap but rather that some stores that are the same in different environments). “Unique city stores” refers to the stores that only belonged to one city (i.e., “Store 4”, “Store 8”, and “Store 9”); “Two shared city stores” refers to stores that only belong to two cities at the same time (i.e., “Store 2”, “Store 6” and “Store 7”); “Three shared city stores” refers to stores

that belong to three cities at the same time (i.e., “Store 1”, “Store 3” and “Store 5”). **c** The three environments shared the same physical layout of stores although the identities of stores differed. **d** An example of an encoding trial from a first-person perspective. Participants encoded the locations of stores and the duration from the center store to peripheral stores. **e** Retrieval consisted of 6 runs of city-specific spatial distance judgments and 6 runs of city-specific temporal durations judgments. After the retrieval task, participants were asked to complete a localizer task involving a vowel counting task.

I had some concerns about how statistics were reported and in some cases computed. It may be a matter of simple clarification in many cases.

R3.2 “Firstly, I think it is important for the authors to report coordinates from their hippocampal searchlight clusters, which will allow readers to relate these outcomes to prior work and anatomical boundaries.--I was confused and concerned about how significance testing was reported - the Methods only indicate a cluster-based significance threshold of $p < 0.01$ based on permutation testing - but at what voxel-wise threshold? Was this $p < 0.05$ relative to 33% classification, for example, for the first hippocampal result reported, being a 3-env classifier? There has been some very lively discussion in the field in recent years, and strong skepticism, for applying cluster-based correction of this sort when using such a liberal height p/t threshold in unsmoothed fMRI data, where assumptions of cluster correction break down. A cluster of just 25 voxels, identified hi-res whole-brain at $p < 0.05$ classification significance, would fall deep into firing zone of those arguments, I think. The authors should clarify, and if this is indeed the case should say more about the reliability of these results in light of that concern.

--Similarly, I found the references to Bonferroni correction in the same early results on hippocampal coding (lines 238+) unclear. Did the authors use the peak value from the cluster identified in the searchlight? Or perhaps the average? What was the Bonferroni correction applied against (from the t-values it is clearly not the whole-brain voxel count, but I was otherwise unsure)? The answers to these questions would also help me appreciate the extent to which this analysis avoids circularity.”

Thank you for the suggestions on the statistics. In the revision, also following the suggestion of reviewer #2 (please also see our detailed response in R2.1), we have replaced the searchlight-based SVM classification analysis with a hippocampal subfield-based SVM classification analysis. This new analysis showed that the overall classification accuracy (across all cities) was above chance level in CA1 ($t(26) = 3.222$, $p = 0.003$, chance level = 33.33%, Fig.5a, please see below). Two-tailed t-tests revealed that this classification was above chance or all cities ($t(26) > 2.100$, $P_s < 0.045$, two-tailed, survived by FDR correction, Fig.5b, please see below).

Fig.5

Environment-specific representations in CA1. **a** The SVM classifier revealed overall classification accuracy in CA1 (across all cities) was well above chance ($t(26) = 3.222$, $p = 0.003$, chance level = 33.33%). **b** Testing each city's classifier performance against chance level in CA1 revealed that the classifier performed above chance on all cities ($t(26) > 2.100$, $P_s < 0.045$, two-tailed, survived by FDR correction). * $p < 0.05$, ** $p < 0.01$.

R3.3 “Why were such tests conducted one-tailed, given that classification can be below chance?”

Thank you for pointing out this. All t-tests in this revision are now two-tailed.

R3.4 “Similar concerns applied to the RIFG and Frontal Pole in terms of significance and circularity in follow-up tests (e.g., line 380+)”

We apologize for not making our analysis clearer in the original submission. In the revision, we correlated the activation of each condition (unique city store within-city PS / unique city store between-city PS) in each voxel of the whole brain during retrieval with the corresponding PS in CA2/3/DG, while controlling for the activation level of hippocampal subfields. The partial correlation coefficients were transformed into Fisher's z-scores and we compared the differences between unique store within-city PS vs. unique store between-city PS. Then, a random-effects model was used for group analysis using a cluster-based threshold of $Z > 3.1$, with $p < 0.05$ (corrected for family-wise error rate using random field theory, see Methods). This revealed a significant cluster located in LIFG (MNI: -52, 22, -2, $Z = 3.91$; please see figure below). Then, we extracted the averaged partial correlation coefficients for the three conditions from the LIFG cluster (voxel number = 99) and compared the correlation coefficients against zero. We attempted to be cautious about the potential for circular analyses so in this t-test analysis, we focused on whether the correlation between activation of levels of LIFG and PS in CA2/3/DG was significantly higher or lower than zero. The positive correlation coefficients suggested the frontal cortex might play a role, through functional

interactions, in enhancing the fidelity of the same city's representation. To avoid circularity, we did not compare between the two conditions (the unique city store within-city PS vs. unique city store between-city PS) which we used to define the LIFG cluster. We've revised this part both in Methods and Results to make it clearer.

R3.4 "My final statistical concern related to the idea of "representational repulsion". On the one hand, it was interesting and encouraging to see some replication and extension of this idea from the recent work from Chanales et al. On the other hand, it is not clear to me that "repulsion" is mechanistically distinct from orthogonal - particularly at the granular level of fMRI, where it is not possible to measure the firing patterns of specific neurons. The authors make a strong distinction there in the manuscript, but my concern rests on two considerations: 1) raw similarity scores are notoriously influenced by tSNR and univariate signal among other things, and so interpretation of a raw magnitude, rather than relative, metric of representation is suspect. 2) although the change in sign seems less ambiguous in meaning, I'm not sure it is, at least at this level of measurement. In traditional RSA work, r is taken as a continuous index of distance from one representation to another and a negative correlation is simply greater distance in a distance matrix and this broad approach can have notable correspondence to neural data (e.g., Kriegeskorte ...Bandettini, 2008, Neuron). If the authors feel strongly that an important change happens from neural patterns approaching orthogonal to passing it (an idea I do find interesting), it feels some theory and methods work may be needed to flesh that out and bolster the argument that it should be interpreted that way."

We thank the reviewer for raising this issue. First, we agree that the raw pattern similarity values could potentially be influenced by tSNR and univariate activation levels. Although we attempted to control for this in our analyses, we agree an additional metric is helpful to ensure convergent findings. Therefore, we calculated a relative metric of representation, the difference of between-city PS between the three shared city trials condition and unique city trials, as a new index for any "repulsion effect". This again revealed a repulsion effect in CA1 ($t(26) = -3.364$, $p = 0.002$, compared to zero, two-tailed) but not in CA2/3/DG ($t(26) = -0.755$, $p = 0.457$). Second, we agree that, as a continuous index of distance, smaller r values past the point of orthogonalization represent greater distance between two matrixes. The repulsion effect that we found in

this study is consistent with theoretical models suggesting that CA1 may represent changes in input in a linear fashion¹³, by which its representational space/range is larger than CA2/3/DG. From this point of view, the mechanism for mediating memory interference in CA1 and CA2/3/DG could be the same but varying in ranges of representation: CA1 could represent overlapped / interfering representations in an exaggerated way compared CA2/3/DG. We have added the details of the results reported above and some more consideration of these issues within the discussion. Please also see the response to R1.2 and R1.3.

Added to the Results:

“The different pattern between CA1 (i.e., repulsion) and CA2/3/DG (i.e., pattern separation) was confirmed by a significant ROI (i.e., CA1, CA2/3/DG) by condition (between-city PS for unique city trials, between-city PS for three shared city trials) interaction effect ($F(1,26) = 4.416$, $p = 0.045$). Importantly, there was no main effect between the two ROIs ($F(1,26) = 0.545$, $p = 0.467$), and the observed repulsion effect in CA1 (but not in CA2/3/DG) could not be accounted by higher between-city PS for unique city trials in CA1 because there was no significant difference for between-city PS for unique city trials between CA1 and CA2/3/DG ($t(26) = 1.800$, $p = 0.117$, two-tailed).”

“Furthermore, considering that raw PS scores can be influenced by tSNR and univariate activation levels, we computed a relative difference score for between-city PS for three shared city trials compared to unique city trials (Fig.3e). This result again revealed a repulsion effect in CA1 ($t(26) = -3.364$, $p = 0.002$, compared to zero, two-tailed), but not in CA2/3/DG ($t(26) = -0.755$, $p = 0.457$, compared to zero, two-tailed).

“Together, these results indicate that both CA2/3/DG and CA1 contributed to the discrimination of overlapping memories but through potentially different mechanisms.”

Added to the Discussion:

“According to this account, positive, zero, and negative MPS correlations are simply different degrees of similarity existing along a continuous spectrum of pattern separation/completion.”

“Such reverse representations have been reported previously for overlapping routes during virtual navigation using fMRI¹¹ although, to the best of our knowledge, not in any previous rodent single neuron studies.”

R3.5 "I found the frontal pole results quite novel, as this area has received little mechanistic attention in navigation work. Given evidence here for this region's relationship to enhancing within-env fidelity and minimizing cross-env interference, I was surprised the authors didn't cite some of the sparse work tying this brain area to replanning+strategization surrounding specific locations/landmarks at retrieval (as reviewed by Spiers and Gilbert, 2015 Frontiers; see also a new review to just come out: Patai ... Spiers 2021 TiCS), and specific examples linking the area

to landmark/goal representation in the hippocampus and to route planning (Brown ... Wagner, 2016 + 2020 Science and Curr Bio)”

We appreciate the reviewer for providing this helpful recommendation. Unfortunately, after applying the partial correlation analysis between the activation of each condition (unique city store within-city PS / unique city store between-city PS) in each voxel of the whole brain during retrieval and the corresponding PS in CA1, we could not replicate the significant result in the frontal pole on our larger sample. Considering this effect within the frontal pole in our original analysis was also not particularly powerful (i.e., it only existed when a lower threshold of $Z = 2.0$), we removed these results from the revision.

In the revision, we correlated the activation of each condition (unique city store within-city PS / unique city store between-city PS) in each voxel of the whole brain during retrieval with the corresponding PS in CA2/3/DG, while controlling for the activation level of hippocampal subfields. The partial correlation coefficients were transformed into Fisher's z-scores and we compared the differences between unique store within-city PS vs. unique store between-city PS. Then, a random-effects model was used for group analysis using a cluster-based threshold of $Z > 3.1$, with $p < 0.05$ (corrected for family-wise error rate using random field theory, see Methods) and a significant cluster located in LIFG (MNI: -52, 22, -2, $Z = 3.91$) was found (please also see response in R3.4).

With regard to the paper mentioned on PFC, we apologize for the oversight in citations and discussion of relevant work. We thank the reviewer for the recommended papers and have added those to the discussion where relevant.

R3.6 “I don't think the authors should summarize and refer to their FPC results as dorsolateral PFC – these areas are different in cytoarchitecture, connectivity, and functional associations in prominent theories of PFC organization (e.g., Badre and Nee, 2018, TiCS; Nee and D'Esposito, 2016, eLife) and it may confuse the literature to do so – depending on where the coordinates fall at least (although this raised for me some uncertainty on where the clusters really fall? Could the authors clarify in revision?)”

We appreciate the reviewer's helpful recommendation. We have removed the results on FPC. Please also see the R3.5

R3.7 "In contrast to the IFG+FPC outcomes which were identified voxel-wise, the authors report outcomes from a very large region of mPFC. One concern is that the underlying functional anatomy is highly heterogeneous (e.g., subdivisions of this mPFC ROI have been tied to goal distance signals and goal-specific coding in hippocampus by various groups; other parts of this ROI, as depicted, encompass anterior cingulate cortex; parts with strong hippocampal connectivity and parts with virtual no direct connectivity with the hippocampus; etc). Another concern is that with more voxels, the ability to detect correlations may be more robust, and it is somewhat challenging to contrast this outcome with those in other prefrontal areas. Although I

would advocate a more precise ROI, I would at least encourage a searchlight analysis to help understand what anatomical loci underlie the overall outcome from this very large region.”

This point is well taken and we thank the reviewer for this helpful suggestion. We agree that the subregions of mPFC are functionally heterogeneous. In response to this concern, we have redone the leave-one-city-out cross-validation SVR classification analysis using a searchlight approach to examine the schematic spatial layout representations in mPFC, which we have now performed across the whole brain. The SVR classifier revealed two clusters whose spatial distances generalized from training on two cities to a third one. This included a more superior cluster (mostly located in paracingulate gyrus, MNI: -4, 50, 18, $Z = 3.68$, Fig.4a, please see below) and a more ventral cluster (mostly located in the anterior cingulate gyrus and paracingulate gyrus, MNI: -10, 34, -4, $Z = 4.02$, Fig.4c, please see below) in mPFC. We have added these new results to the Results and Methods.

Representation of the schematic spatial layout information across environments in medial PFC. **a** The searchlight leave-one-city-out SVR classification revealed a more superior cluster (MNI: -4, 50, 18, $Z = 3.68$) in mPFC whose spatial distances generalized from training on two cities to a third one. **b** The SVR performance for each city in superior mPFC revealed above chance performance on cities 1, 2 and 3. **c** The searchlight leave-one-city-out SVR classification revealed a more ventral cluster (MNI: -10, 34, -4, $Z = 4.02$) in mPFC whose spatial distances

generalize from training on two cities to a third one. **d** The SVR performance for each city in ventral mPFC revealed above chance performance on cities 1, 2 and 3. **p < 0.01, ***p < 0.001.

References

- 1 Ekstrom, A., Suthana, N., Millett, D., Fried, I. & Bookheimer, S. Correlation Between BOLD fMRI and Theta-Band Local Field Potentials in the Human Hippocampal Area. *Journal of Neurophysiology* **101**, 2668-2678, doi:10.1152/jn.91252.2008 (2009).
- 2 Ekstrom, A. How and when the fMRI BOLD signal relates to underlying neural activity: the danger in dissociation. *Brain Res Rev* **62**, 233-244, doi:10.1016/j.brainresrev.2009.12.004 (2010).
- 3 Ekstrom, A. D. Regional variation in neurovascular coupling and why we still lack a Rosetta Stone. *Philos Trans R Soc Lond B Biol Sci* **376**, 20190634, doi:10.1098/rstb.2019.0634 (2021).
- 4 Kyle, C. T., Stokes, J. D., Lieberman, J. S., Hassan, A. S. & Ekstrom, A. D. Successful retrieval of competing spatial environments in humans involves hippocampal pattern separation mechanisms. *Elife* **4**, e10499, doi:10.7554/eLife.10499 (2015).
- 5 Doeller, C. F., Barry, C. & Burgess, N. Evidence for grid cells in a human memory network. *Nature* **463**, 657-661, doi:10.1038/nature08704 (2010).
- 6 Ekstrom, A. D., Harootyan, S. K. & Huffman, D. J. Grid coding, spatial representation, and navigation: Should we assume an isomorphism? *Hippocampus* **30**, 422-432, doi:<https://doi.org/10.1002/hipo.23175> (2020).
- 7 Ekstrom, A. D. Regional variation in neurovascular coupling and why we still lack a Rosetta Stone. *Philosophical Transactions of the Royal Society B* **376**, 20190634 (2021).
- 8 Kriegeskorte, N. Relating population-code representations between man, monkey, and computational models. *Frontiers in Neuroscience* **3**, 35 (2009).
- 9 Muller, R. U., Kubie, J. L., Bostock, E., Taube, J. & Quirk, G. Spatial firing correlates of neurons in the hippocampal formation of freely moving rats. (1991).
- 10 Colgin, L. L., Moser, E. I. & Moser, M.-B. Understanding memory through hippocampal remapping. *Trends in Neurosciences* **31**, 469-477, doi:<https://doi.org/10.1016/j.tins.2008.06.008> (2008).
- 11 Chanales, A. J. H., Oza, A., Favila, S. E. & Kuhl, B. A. Overlap among Spatial Memories Triggers Repulsion of Hippocampal Representations. *Curr Biol* **27**, 2307-2317 e2305, doi:10.1016/j.cub.2017.06.057 (2017).
- 12 Chanales, A. J. H., Tremblay-McGaw, A. G. & Kuhl, B. A. Adaptive repulsion of long-term memory representations is triggered by event similarity. doi:10.1101/2020.01.14.900381 (2020).
- 13 Guzowski, J. F., Knierim, J. J. & Moser, E. I. Ensemble dynamics of hippocampal regions CA3 and CA1. *Neuron* **44**, 581-584, doi:10.1016/j.neuron.2004.11.003 (2004).
- 14 Badre, D. & Wagner, A. D. Left ventrolateral prefrontal cortex and the cognitive control of memory. *Neuropsychologia* **45**, 2883-2901, doi:<https://doi.org/10.1016/j.neuropsychologia.2007.06.015> (2007).

REVIEWER COMMENTS

Reviewer #1 (Remarks to the Author):

I again feel that overall this is a very nice experiment and set of findings. However, I feel that the fundamental issues I raised in how the paper is being framed have not been well-addressed by the small changes to the text. Rest assured, I know the authors are aware of a number of the concerns I raised and their prior record on it. That is, in part, why I phrased it the way I did in my original review. To this reviewer, they are making a logical error atypical of them. Perhaps I have missed a key step in the logical chain and that this could be restructured or clarified. But, if I'm missing it, chances are other readers would as well. The entirety of the rest of this review can be summarized by saying that, despite their protestations to the contrary in their response (where they say they are restricting themselves to the "information coded by the neuron activity"), they are using fMRI patterns to make conclusions about unit-level representations. The theoretical setup for the analyses and the conclusions drawn are based on this and, without an explicit framing of the results at the outset to guide the reader as to what can and cannot be concluded here, there is a real potential for over-interpretation. I understand that this review is quite long for making one basic point, but I wanted to attempt to be clearer than I had in my prior review.

I feel the authors are still mixing voxel-wise and neuron-wise codes in their arguments and the role of "reversed" patterns is still unclear. The specific sentences I pointed out have changed, but the core issue remains (or I am missing a part of their argument). On p 4-5, they lay out the argument for "reversed" codes that move "past the point of orthogonalization" and motivate this on the computational ideas conceptually at the unit level. The key aspect of their argument is that "According to this mechanism, the similarity of overlapping sections between Supermarket A and B would be repulsed to a greater extent, exhibiting reverse similarity between overlapping sections compared to the similarity of two different sections (i.e., similarity between two food sections should be lower than the similarity between the food sections and the parking lot)." So, we are talking about codes at the level of neural representations (the same kinds of codes used to motivate pattern separation in the first place) and they are making a computational argument that such reversed codes will critically have less similarity for the originally more-similar two food sections than for a random pairing (food section to parking lot). The argument here is clearly one based on actual neural codes, yet the data used to support this come only from fMRI studies which, the authors point out in their reply, are more about "information" than about actual neural codes. If such reversed / repulsive codes exist at the neural level and at the fMRI level, that would be one thing. But, to bring up a neural, computational argument and support it purely based on fMRI is making the assumption that the fMRI evidence of "reversal" can be used to infer something about underlying neural reversal. This was the point I was trying to make in my prior review that has not been grappled with adequately.

While on this point, however, there is another key issue. Computationally, it's not clear how being "past the point of orthogonalization" is actually "more dissimilar", especially if one is making the argument based on information content. Two random vectors will have some amount of correlation, though with sparse codes, it will be quite low. They won't be perfectly orthogonal and one might argue for a mechanism that enforces true orthogonality in the example given above. But, they're arguing for a "reverse code" as being more effective. The trouble is that a real "reverse code" has the same exact information as the original code. Let's take their supermarket example. Suppose in Supermarket A, when I walk in, the dairy is on the left and the produce is on the right and I further know that within the dairy, the eggs are in the front of the store, the milk at the back and within the produce the fruits are in the front and the vegetables are in the back. We might code eggs then as $[-1,-1]$, milk as $[-1,+1]$, fruits as $[+1,-1]$ and vegetables as $[+1,+1]$. In an orthogonal code, knowing the representation for Supermarket A won't predict the representation in Supermarket B. But, in a reversed code, all we need to do is invert A's code. So eggs are $[+1,+1]$ – right side, back of store. I can know the full layout of all of Supermarket B by knowing A and simply flipping the code as the vectors simply point in the opposite direction. "Head 10 steps NNE" is the same as "head -10 steps SSW". Knowing the authors' work, it's quite likely that they have something else in mind here concerning the power of these reversed codes in this situation. But, depending on exactly how it is implemented, computationally, changing the connections in a reverse code may well change the same connections being used in the normal code, leading to the exact kind of catastrophic interference that pattern separation was meant

to avoid. But, if these reverse codes are not neural codes per se and come about based on how fMRI is reflecting the neural activity, this concern of mine may not apply. Yet, in this paragraph, they are mixing computational / neural code arguments, reversal codes purely from fMRI, and the notion that such reversed codes may be “more different” than entirely orthogonal codes. I hope they can clarify the picture here for readers like myself.

To be concrete, based on their reply (and their prior work), it is clear that they perfectly understand the concept of pattern separation at the neural level. “When rodents are put in two different environments, the correlation between the firing activity of place cells in the two environments is near zero, termed “remapping,” and is likely related to pattern separation⁹. Previous rodent studies found that even when two environments were highly similar, place cells fired in distinct ways, but not in a negative relationship¹⁰.” It is also perfectly clear that they understand “the lack of a direct mapping between hippocampal BOLD and single neuron activity”. So therefore, it should be clear that we can have neural pattern separation that would reflect in no change in BOLD fMRI multivariate activity patterns. It could also, of course, be the case that, depending on larger-scale representations, neural pattern separation would lead to a reduction or even orthogonalization of fMRI representations. With that in place, it’s hardly a leap to believe that neural pattern separation might, in fact, lead to BOLD “reversal” patterns that are “more than orthogonal”, given the massive amount of summation across neurons, the complex relationship with synaptic activity, etc. The observation of BOLD “reversal” patterns may have no basis for any reversal of patterns at the neural level (and may reflect normal orthogonalization), based on their reply and prior work. If they wish to stick at the level of fMRI representations and information that can be extracted from that, even the computational advantages that pattern separation offers don’t really apply as that underlying separation could lead to such a wide range of fMRI pattern changes. We can’t both acknowledge these limitations of BOLD fMRI and the representational level it works at and, at the same time, use its patterns to infer something at the computational level that’s based on unit-level descriptions the way they do in the Introduction.

In my prior review, I pointed out a few spots where this issue of the fMRI vs neural codes appeared conflated and I just spent far too much text on it above. While they have modified the text in some places to help here, they’ve missed other parts and have added new ones. Take, for example the new text on p. 7 that reads “We sought to test whether neural codes within the hippocampus are holistic”. I have no way to read that other than their goal being to test patterns of activity across neurons within the hippocampus – i.e., not an fMRI abstraction of information, but actual firing rate patterns.

Likewise on p 11 “Earlier, we reported that the neural representations of stores shared between multiple cities were significantly less correlated than stores that were unique to the current city being retrieved”, the phrase “neural representations” makes a similar implication, at least when there has been nothing in the Introduction to setup their approach and its limitations but to guide the reader what they mean by terms like “neural representations”.

Also on p. 11, they have “Alternatively, if the memory representation of the shared city trials is orthogonal between cities (i.e., due to pattern separation), the between-city PS of shared city trials should be comparable (statistically equivalent) to the between-city PS of unique-city trials (Fig.3a, middle panel).” Again, they certainly seem to be talking about inferring unit-level orthogonality based on fMRI patterns.

On page 18, “Accumulating evidence suggests that hippocampal pattern separation may serve as an important means for distinguishing similar stimuli¹²⁻¹⁴, with our findings suggesting that this computational mechanism may play a partial role in distinguishing shared elements across cities. ... Particularly in our experimental design, the shared city trials involved identical stores and were shared between two or three cities. Therefore, pattern separation in CA2/3/DG may be important to forming distinct codes for these competing trials to reduce memory interference and allow successful retrieval of environmental information.” This again is blending between actual unit-level neural codes and computations and the fMRI patterns.

In summary, they have a solid experiment and analysis path. But, it is my opinion that at the outset of the paper, they need to give the reader a better understanding of what fMRI patterns will be able to do in improving our understanding of memory and memory representations rather than lead them down the path of thinking about unit-level neural representations. It’s not merely adjusting a sentence or two in the Discussion, but it’s certainly something that can be done that will lead to a stronger paper.

Reviewer #2 (Remarks to the Author):

The revision by Zheng and colleagues presents a comprehensive and successful reply to the reviewers' initial critiques. In particular, the revised main text has been organized to better represent the logical connection between analyses and the rationale of the different techniques employed is well described. Notably, the expanded sample size greatly strengthen the manuscript. Although this may remain behind the opaque curtain of review, it is worth noting that the central findings do remain when including such a large proportion of additional participants and speaks to the strength of the effects. Also, conducting additional data collection during COVID times must have been a challenging task for which the authors should be applauded. Finally, the revised descriptions of methods and approach clear up my initial confusions. I have one comment below, but feel that this revised manuscript better represents the findings and offers a valuable contribution to our understanding of hippocampal- and PFC-related reinstatement during spatial navigation.

In my original review, I questioned the rationale for choosing the SVR approach versus perhaps the more conventional MPS analysis in looking for the PFC schema representations. First of all, I appreciate the newly added MPS analysis and think that the converging results help bridge the gap to the SVR findings. However, I think it is perhaps an interesting point that the MPS findings are overall weaker than the SVR findings. Perhaps a quick discussion of the relative strengths and weakness of the two approaches, especially in terms of their analytic power, would be warranted. Guidance for choosing SVR would be helpful for any motivated reader looking to implement a similar analysis.

Reviewer #3 (Remarks to the Author):

I would like to thank the authors for their thorough and considerate response to my concerns. Although some of the results ultimately changed in the course of revision, I believe the manuscript is now stronger and the core outcomes and interpretations are very well argued. This will make a very nice contribution to the field

Reviewer #1 (Remarks to the Author):

R1.1 I again feel that overall this is a very nice experiment and set of findings. However, I feel that the fundamental issues I raised in how the paper is being framed have not been well-addressed by the small changes to the text. Rest assured, I know the authors are aware of a number of the concerns I raised and their prior record on it. That is, in part, why I phrased it the way I did in my original review. To this reviewer, they are making a logical error atypical of them. Perhaps I have missed a key step in the logical chain and that this could be restructured or clarified. But, if I'm missing it, chances are other readers would as well. The entirety of the rest of this review can be summarized by saying that, despite their protestations to the contrary in their response (where they say they are restricting themselves to the "information coded by the neuron activity"), they are using fMRI patterns to make conclusions about unit-level representations. The theoretical setup for the analyses and the conclusions drawn are based on this and, without an explicit framing of the results at the outset to guide the reader as to what can and cannot be concluded here, there is a real potential for over-interpretation. I understand that this review is quite long for making one basic point, but I wanted to attempt to be clearer than I had in my prior review.

We thank the reviewer for the positive feedback on the design and implementation of our experiment reported here, and we apologize if the reviewer felt that we had not addressed the issue of fMRI vs. neural signals in sufficient depth. Here, we try to explain the rationale and logic in detail behind our experiment with regard to fMRI and make changes to the manuscript to reflect this logic and rationale.

As we are confident the reviewer knows, unlike animal studies, which often involve recording single neuron activity directly, human studies often employ fMRI to measure hemodynamic signals as an indirect proxy for neural activity. A key point we would like to emphasize, that perhaps we were not sufficiently clear about in the last revision, is that while increases and decreases in small numbers of recorded single neurons / local field potentials at individual electrodes show only a weak¹ correspondence, if any at all², with fMRI in the human hippocampus, multivariate approaches to neural signals have revealed reinstatement-related codes in the hippocampus for all three measures³⁻⁸ during episodic memory tasks. While this does not mean they reflect identical signatures, there are also some studies showing a more direct correspondence between multivariate fMRI BOLD activity pattern and populations of single neurons activity pattern in brain regions outside of the hippocampus⁹⁻¹¹. In other words, we think past work makes it reasonable to hypothesize that distributed patterns of single neuron or LFP activity could be reflected in the distributed BOLD signals. While this statement remains to be directly tested in the human hippocampus,

we have performed simulations to test this idea, which we detail below, at least affirming this possibility.

We agree though that there is a potential logical disconnect that needs clarifying here: while pattern separation mechanisms were first demonstrated using single neuron recordings in rodents¹², and such codes have been detected using both distributed local field potential recordings in humans and fMRI¹³⁻¹⁵, repulsion-like codes have only been demonstrated using fMRI^{16,17}. In this way, it is an assumption, (possibly a strong one), that repulsion-like codes are reflected at the single neuron level. In addition, we agree that it remains somewhat of an assumption that pattern separation activity detected with single neuron recordings in rodents is reflected in either the local field potential or distributed voxel codes using fMRI, although not an unreasonable one, in our view.

To deal with these issues, we have made changes to the manuscript to better reflect that we are employing studies from rodents (and in some cases, human invasive studies) to generate novel hypotheses about how environmental knowledge might be stored in the human brain. We have also attempted to be clearer within the manuscript what is known about studies comparing univariate (increases and decreases in BOLD) vs. multivariate distributed codes. Finally, we have performed detailed simulations showing that distributed changes in either the patterns of local field potentials or single neural activity, provided such signals are sufficiently distributed, can be reflected in similar distributed changes using MVPA at the level of fMRI.

The last issue we would like to clarify, which in part relates to the issue of what the BOLD signal means, is whether it is justified to even call fMRI “neural.” While changes in BOLD can have both glial and neural origins via signaling to the vasculature¹⁸⁻²⁰, we think it is clear that changes in neural activity are driving at least some of the variance in changes in the BOLD signal, particularly when looking at the population rather than single voxel level²¹. This is supported by several studies that have shown a striking correspondence between the patterns of activity across a region of interest using fMRI and those recorded with single neurons⁹⁻¹¹. This is an important point and we believe justifies our statement regarding “neural” codes within the hippocampus. Nonetheless, we have attempted to be clearer about what is known and what is not known and how single neuron activity helps us generate novel hypotheses we can test with fMRI rather than assuming a direct relationship between the two.

Regarding the novel simulation we have performed, which we have now included in Supplementary Note 7, we tested whether distributed changes in neural activity (as reflected at the level either of single neurons or the LFP) could, in principle, reflect at the level of distributed changes with fMRI. The key assumption here, which we remain agnostic about but think is reasonable, is that the *summed* activity across thousands of

neurons does reflect, in some form, in the BOLD signal (i.e., the correlation for summed activity across thousands of neurons and changes at a single voxel are not random). Note that past studies have not employed sufficient numbers of recording electrodes to detect all (or even an appreciable fraction of neurons) within a voxel and thus the issue remains untested. However, even with a sparse coding scheme, such as observed for both neurons and vasculature within the hippocampus²¹, we think that this assumption would still likely hold as a large enough collection of neurons would surely elicit blood flow changes that can deliver metabolic energy. In our simulation, we created random matrices that either showed zero correlation (e.g., orthogonal), a positive correlation (e.g., pattern similarity), or a negative correlation (e.g., repulsion). We then binned and summed over putative voxels. Across thousands of simulations, we found a consistent and strong correspondence between zero, positive, and negative distributed patterns for simulated neural and fMRI patterns. In our example, we flipped elements of the matrix to ones to create such correlations but we also tested other values (zeros, random distributions centered around zero) and got identical results.

Newly included material to address fMRI and electrophysiology (introduction)

“Single neuron studies demonstrate that place cells “remapping” in which place cell firing patterns show a near zero correlation between two (in some cases similar) environments, could relate to such pattern separation mechanisms²². Recently, by utilizing multivariate pattern analysis (MVPA) on a population of neurons or local field potential (LFP) in hippocampus, researchers found that both the pattern of neural firing rates²³ or LFP signals¹³ could support such putative pattern separation mechanisms. Here, we leverage MVPA and high-resolution fMRI to test novel hypotheses derived from past single neuron²³, LFP¹³, and fMRI studies^{14,15} regarding how pattern separation might relate to the issue of environment-specific codes through mechanisms by which inputs that share some similarities – like supermarkets – can be stored separately.”

“We leveraged these past findings related to repulsion to test novel hypotheses based on the assumption that “reversed” pattern similarity – past the point orthogonalization could be a potential mechanism for maintaining overlapping representations.”

Results

“We sought to test whether distributed neural codes within the hippocampus might reveal evidence for multivariate codes related to holistic retrieval.”

“Earlier, we reported that the distributed patterns of neural activity for stores shared between multiple cities were significantly less correlated than stores that were unique to the current city being retrieved”

“Alternatively, if the memory representation of the shared city trials is orthogonal between cities (perhaps related to pattern separation), the between-city PS of shared city trials should be comparable (statistically equivalent) to the between-city PS of unique-city trials (Fig.3a, middle panel).”

Discussion

“As we noted in the introduction, however, one issue with holistic retrieval is that it does not provide an obvious means for distinguishing shared elements across environments. Accumulating evidence suggests that pattern separation within the hippocampus may serve as an important means for distinguishing similar stimuli²⁴⁻²⁶, with our findings suggesting such a mechanism, at least as detected using MVPA and high-resolution fMRI, could play a partial role in distinguishing shared elements across cities.”

“Particularly in our experimental design, the shared city trials involved identical stores and were shared between two or three cities. Therefore, reduction in the distributed patterns of neural activity in CA2/3/DG may be important to forming distinct codes for these competing trials to reduce memory interference and allow successful retrieval of environmental information.”

“Past studies have shown pattern separation mechanisms in the hippocampus using single neuron activity^{12,22}, multi-neuron activity patterns²³, LFP activity patterns¹³, and fMRI activity pattern¹⁴, and while only past fMRI studies have shown repulsion^{16,17}, we note that theoretically such a mechanisms could represent a viable, although untested, manner by which single neuron activity and LFPs could also process overlapping information. Such reverse representations have been reported previously for overlapping routes during virtual navigation using fMRI¹⁶ although, to the best of our knowledge, not in any previous rodent single neuron studies or fMRI studies requiring environment-specific retrieval.”

R1.2 *I feel the authors are still mixing voxel-wise and neuron-wise codes in their arguments and the role of “reversed” patterns is still unclear. The specific sentences I pointed out have changed, but the core issue remains (or I am missing a part of their argument). On p 4-5, they lay out the argument for “reversed” codes that move “past the point of orthogonalization” and motivate this on the computational ideas conceptually at the unit level. The key aspect of their argument is that “According to this mechanism, the similarity of overlapping sections between Supermarket A and B would be repulsed to a greater extent, exhibiting reverse similarity between overlapping sections compared to the similarity of two different sections (i.e., similarity between two food sections should be lower than the similarity between the food sections and the parking lot).” So,*

we are talking about codes at the level of neural representations (the same kinds of codes used to motivate pattern separation in the first place) and they are making a computational argument that such reversed codes will critically have less similarity for the originally more-similar two food sections than for a random pairing (food section to parking lot). The argument here is clearly one based on actual neural codes, yet the data used to support this come only from fMRI studies which, the authors point out in their reply, are more about “information” than about actual neural codes. If such reversed / repulsive codes exist at the neural level and at the fMRI level, that would be one thing. But, to bring up a neural, computational argument and support it purely based on fMRI is making the assumption that the fMRI evidence of “reversal” can be used to infer something about underlying neural reversal. This was the point I was trying to make in my prior review that has not been grappled with adequately.

While on this point, however, there is another key issue. Computationally, it's not clear how being “past the point of orthogonalization” is actually “more dissimilar”, especially if one is making the argument based on information content. Two random vectors will have some amount of correlation, though with sparse codes, it will be quite low. They won't be perfectly orthogonal and one might argue for a mechanism that enforces true orthogonality in the example given above. But, they're arguing for a “reverse code” as being more effective. The trouble is that a real “reverse code” has the same exact information as the original code. Let's take their supermarket example. Suppose in Supermarket A, when I walk in, the dairy is on the left and the produce is on the right and I further know that within the dairy, the eggs are in the front of the store, the milk at the back and within the produce the fruits are in the front and the vegetables are in the back. We might code eggs then as $[-1, -1]$, milk as $[-1, +1]$, fruits as $[+1, -1]$ and vegetables as $[+1, +1]$. In an orthogonal code, knowing the representation for Supermarket A won't predict the representation in Supermarket B. But, in a reversed code, all we need to do is invert A's code. So eggs are $[+1, +1]$ – right side, back of store. I can know the full layout of all of Supermarket B by knowing A and simply flipping the code as the vectors simply point in the opposite direction. “Head 10 steps NNE” is the same as “head -10 steps SSW”. Knowing the authors' work, it's quite likely that they have something else in mind here concerning the power of these reversed codes in this situation. But, depending on exactly how it is implemented, computationally, changing the connections in a reverse code may well change the same connections being used in the normal code, leading to the exact kind of catastrophic interference that pattern separation was meant to avoid. But, if these reverse codes are not neural codes per se and come about based on how fMRI is reflecting the neural activity, this concern of mine may not apply. Yet, in this paragraph, they are mixing computational / neural code arguments, reversal codes purely from fMRI, and the notion that such reversed codes

may be “more different” than entirely orthogonal codes. I hope they can clarify the picture here for readers like myself.

We appreciate the point about “reversed” codes and repulsion. While we agree that no single neuron evidence supports the idea of “reversed” codes, given that pattern separation also works at the population-level, and past studies have shown evidence for repulsion mechanisms with fMRI at the distributed population level^{16,17}, we think it is reasonable to hypothesize that such “reversed” neural codes could also be detected using the distributed BOLD signal during learning of competing spatial environments. But we do not intend to imply that the observed “reversed” codes in the current study must correspond with “reversed” distributed single neuron codes and have attempted to clarify that point in more detail. They could correspond to the local field potential or something entirely different, although we think our simulation provides at least tentative support for the idea that the multivariate patterns of fMRI changes we observed could relate to either distributed single neuron or LFP changes. We also agree that “repulsed” mechanisms could, in principle, potentially lead to greater interference within the network, although we think they could also serve as a mechanism for reduction interference as well. To the best of our knowledge, these issues have not been tested with large patterns of single neurons and therefore the question appears to be open.

To clarify in our experiment regarding the reviewer’s specific example, in our experiment, we employed three cities. If the codes for the common stores (i.e., Store 1, Store 3, and Store 5) in City 1 are [1, 1], [3, 3], and [5, 5] respectively, the codes for the common stores in City 2 should be [-1, -1], [-3, -3], and [-5, -5]. In this case, though, there is ambiguity about the codes in City 3 for the common stores. If they are [1, 1], [3, 3] and [5, 5] in City 3, they would be completed with the same with the codes in City 1. This creates an issue, however, in that participants did not confuse City 1 and 3. Indeed, we would like to point out that there is not a one-to-one mapping between a “reversed” code and the physical location of the store (e.g., as measured by a small number of place fields during electrophysiological recordings in rodents). However, what we can get from our data is the representation of the common stores are reversed with each other, not the position of them or the connection between them. In fact, the positions of the shared stores are the same across the three environments and at the same time the neural representations of them are statistically different, which we believe is related to the participant’s ability to discriminate them.

Another potential issue here is that the dimensionality of the vectors (matrices) is likely much higher than $N = 2$. Again, in this way, what we believe that we are picking up on is unlikely to be place cell activity alone but rather something more distributed and possibly a form of global “mixed” selectivity (see also Nolan et al.²⁷). For such a

distributed code, there are likely small changes in elements across millions of elements. In our simulations, importantly, we flipped a relatively small number of elements to obtain negatively and positively correlated matrices. While we understand the reviewer was using a simplified example to make a point, we think it is important to note that we are considering highly distributed codes, which likely have different properties. In sum, while we appreciate the reviewer's point, we believe that his/her speculation is based on a small number of single neuron remappings for each location which we do not believe is the case in our study due to a much larger matrix size and lack of direct correspondence between vector elements and locations.

R1.3 *To be concrete, based on their reply (and their prior work), it is clear that they perfectly understand the concept of pattern separation at the neural level. "When rodents are put in two different environments, the correlation between the firing activity of place cells in the two environments is near zero, termed "remapping," and is likely related to pattern separation⁹. Previous rodent studies found that even when two environments were highly similar, place cells fired in distinct ways, but not in a negative relationship¹⁰." It is also perfectly clear that they understand "the lack of a direct mapping between hippocampal BOLD and single neuron activity". So therefore, it should be clear that we can have neural pattern separation that would reflect in no change in BOLD fMRI multivariate activity patterns. It could also, of course, be the case that, depending on larger-scale representations, neural pattern separation would lead to a reduction or even orthogonalization of fMRI representations. With that in place, it's hardly a leap to believe that neural pattern separation might, in fact, lead to BOLD "reversal" patterns that are "more than orthogonal", given the massive amount of summation across neurons, the complex relationship with synaptic activity, etc. The observation of BOLD "reversal" patterns may have no basis for any reversal of patterns at the neural level (and may reflect normal orthogonalization), based on their reply and prior work. If they wish to stick at the level of fMRI representations and information that can be extracted from that, even the computational advantages that pattern separation offers don't really apply as that underlying separation could lead to such a wide range of fMRI pattern changes. We can't both acknowledge these limitations of BOLD fMRI and the representational level it works at and, at the same time, use its patterns to infer something at the computational level that's based on unit-level descriptions the way they do in the Introduction.*

We appreciate and agree the Reviewer's comments on the interpretation of "reversed" representation. We have added a point to the discussion to address this issue:

“Finally, given the lack of a direct mapping between hippocampal BOLD fMRI and single neuron / local field potential activity recorded at single electrodes in the hippocampus^{2,28}, it is unclear exactly how the fMRI findings here related to pattern separation and repulsion in particular might relate to the activity of place cells. While there is evidence supporting a correspondence between multivariate patterns at the level of fMRI BOLD activity and populations of single neurons / LFPs^{3,4,9-11}, whether a population of single neural / LFP codes exhibit the same reversed codes is still need of direct testing. To more directly address this issue, we performed simulations of how large groups of neurons in displaying distributed codes might relate to distributed patterns of voxel-based activity using fMRI, our simulations support the idea that, under some situations, both “separation” and “repulsion” mechanisms at the level of single neurons / LFPs, provided they are sufficiently distributed, can be detected with MVPA methods at the level of changes in patterns across voxels (Supplementary Note 7).”

R1.4 *In my prior review, I pointed out a few spots where this issue of the fMRI vs neural codes appeared conflated and I just spent far too much text on it above. While they have modified the text in some places to help here, they’ve missed other parts and have added new ones. Take, for example the new text on p. 7 that reads “We sought to test whether neural codes within the hippocampus are holistic”. I have no way to read that other than their goal being to test patterns of activity across neurons within the hippocampus – i.e., not an fMRI abstraction of information, but actual firing rate patterns.*

Likewise on p 11 “Earlier, we reported that the neural representations of stores shared between multiple cities were significantly less correlated than stores that were unique to the current city being retrieved”, the phrase “neural representations” makes a similar implication, at least when there has been nothing in the Introduction to setup their approach and its limitations but to guide the reader what they mean by terms like “neural representations”.

Also on p. 11, they have “Alternatively, if the memory representation of the shared city trials is orthogonal between cities (i.e., due to pattern separation), the between-city PS of shared city trials should be comparable (statistically equivalent) to the between-city PS of unique-city trials (Fig.3a, middle panel).” Again, they certainly seem to be talking about inferring unit-level orthogonality based on fMRI patterns.

On page 18, “Accumulating evidence suggests that hippocampal pattern separation may serve as an important means for distinguishing similar stimuli¹²⁻¹⁴, with our findings suggesting that this computational mechanism may play a partial role in distinguishing shared elements across cities. ... Particularly in our experimental design, the shared city trials involved identical stores and were shared between two or three

cities. Therefore, pattern separation in CA2/3/DG may be important to forming distinct codes for these competing trials to reduce memory interference and allow successful retrieval of environmental information.” This again is blending between actual unit-level neural codes and computations and the fMRI patterns.

In summary, they have a solid experiment and analysis path. But, it is my opinion that at the outset of the paper, they need to give the reader a better understanding of what fMRI patterns will be able to do in improving our understanding of memory and memory representations rather than lead them down the path of thinking about unit-level neural representations. It’s not merely adjusting a sentence or two in the Discussion, but it’s certainly something that can be done that will lead to a stronger paper.

We have attempted, as best as possible, to address the flagged sentences above (please see below). While we think it is reasonable to associate fMRI with “neural” changes, we also agree that it is not reasonable to assume an exact correspondence between single neural changes (e.g., place cells) and fMRI, particularly in the hippocampus. In addition, we have tried to clarify throughout that we are focusing on distributed patterns of activity rather than individual neurons recorded at a small number of electrodes.

For convenience, we have copied the modified sentences below:

“We sought to test whether distributed neural codes within the hippocampus might reveal evidence for multivariate codes related to holistic retrieval.”

“Earlier, we reported that the distributed patterns of neural activity for stores shared between multiple cities were significantly less correlated than stores that were unique to the current city being retrieved.”

“Alternatively, if the memory representation of the shared city trials is orthogonal between cities (perhaps related to pattern separation), the between-city PS of shared city trials should be comparable (statistically equivalent) to the between-city PS of unique-city trials (Fig.3a, middle panel).”

“Accumulating evidence suggests that pattern separation within the hippocampus may serve as an important means for distinguishing similar stimuli²⁴⁻²⁶, with our findings suggesting such a mechanism, at least as detected using MVPA and high-resolution fMRI, could play a partial role in distinguishing shared elements across cities.”

“Particularly in our experimental design, the shared city trials involved identical stores and were shared between two or three cities. Therefore, reduction in the distributed patterns of neural activity in CA2/3/DG may be important to forming distinct codes for

these competing trials to reduce memory interference and allow successful retrieval of environmental information.”

Reviewer #2 (Remarks to the Author):

The revision by Zheng and colleagues presents a comprehensive and successful reply to the reviewers' initial critiques. In particular, the revised main text has been organized to better represent the logical connection between analyses and the rationale of the different techniques employed is well described. Notably, the expanded sample size greatly strengthen the manuscript. Although this may remain behind the opaque curtain of review, it is worth noting that the central findings do remain when including such a large proportion of additional participants and speaks to the strength of the effects. Also, conducting additional data collection during COVID times must have been a challenging task for which the authors should be applauded. Finally, the revised descriptions of methods and approach clear up my initial confusions. I have one comment below, but feel that this revised manuscript better represents the findings and offers a valuable contribution to our understanding of hippocampal- and PFC-related reinstatement during spatial navigation.

In my original review, I questioned the rationale for choosing the SVR approach versus perhaps the more conventional MPS analysis in looking for the PFC schema representations. First of all, I appreciate the newly added MPS analysis and think that the converging results help bridge the gap to the SVR findings. However, I think it is perhaps an interesting point that the MPS findings are overall weaker than the SVR findings. Perhaps a quick discussion of the relative strengths and weakness of the two approaches, especially in terms of their analytic power, would be warranted. Guidance for choosing SVR would be helpful for any motivated reader looking to implement a similar analysis.

We thank the Reviewer for the positive feedback and appreciate the suggestion on the discussion about the comparison between SVR and MPS methods. We have no preference on the two approaches and both of them revealed different aspects of spatial schema in the current experiment design: the SVR classification analysis emphasize on the spatial distances across stores while the MPS analysis focuses on the same location information across three cities. Considering that the experiment requires the participants to compare the spatial distances, it is reasonable to speculate that the representation of spatial distances would be more readily to be decoded by SVR classification analysis.

We have also added the following to the Discussion:

“The leave-one-city out SVR classification analysis places more emphasis on decoding spatial distance information while the MPS approach focused on the same location information across three cities. While both two approaches revealed different aspects of spatial schema representations, the SVR findings may be better positioned than the MPS findings to allow inference about spatial schema because the SVR analysis explicitly utilized distance measures.”

Reviewer #3 (Remarks to the Author):

I would like to thank the authors for their thorough and considerate response to my concerns. Although some of the results ultimately changed in the course of revision, I believe the manuscript is now stronger and the core outcomes and interpretations are very well argued. This will make a very nice contribution to the field

Reference

- 1 Ekstrom, A., Suthana, N., Millett, D., Fried, I. & Bookheimer, S. Correlation Between BOLD fMRI and Theta-Band Local Field Potentials in the Human Hippocampal Area. *J Neurophysiol* **101**, 2668-2678 (2009).
- 2 Hill, P. F. *et al.* Distinct neurophysiological correlates of the fMRI BOLD signal in the hippocampus and neocortex. *Journal of Neuroscience* (2021).
- 3 Manning, J. R., Polyn, S. M., Baltuch, G. H., Litt, B. & Kahana, M. J. Oscillatory patterns in temporal lobe reveal context reinstatement during memory search. *Proc Natl Acad Sci U S A* **108**, 12893-12897, doi:10.1073/pnas.1015174108 (2011).
- 4 Manning, J. R., Sperling, M. R., Sharan, A., Rosenberg, E. A. & Kahana, M. J. Spontaneously reactivated patterns in frontal and temporal lobe predict semantic clustering during memory search. *J Neurosci* **32**, 8871-8878, doi:10.1523/JNEUROSCI.5321-11.2012 (2012).
- 5 Lohnas, L. J. *et al.* Time-resolved neural reinstatement and pattern separation during memory decisions in human hippocampus. *Proceedings of the National Academy of Sciences* **115**, E7418-E7427, doi:10.1073/pnas.1717088115 (2018).
- 6 Tompary, A., Duncan, K. & Davachi, L. High-resolution investigation of memory-specific reinstatement in the hippocampus and perirhinal cortex. *Hippocampus* **26**, 995-1007, doi:<https://doi.org/10.1002/hipo.22582> (2016).
- 7 Pacheco Estefan, D. *et al.* Coordinated representational reinstatement in the human hippocampus and lateral temporal cortex during episodic memory retrieval. *Nature Communications* **10**, 2255, doi:10.1038/s41467-019-09569-0 (2019).
- 8 Miller, J. F. *et al.* Neural activity in human hippocampal formation reveals the spatial context of retrieved memories. *Science* **342**, 1111-1114, doi:10.1126/science.1244056 (2013).
- 9 Kriegeskorte, N. *et al.* Matching categorical object representations in inferior temporal cortex of man and monkey. *Neuron* **60**, 1126-1141, doi:10.1016/j.neuron.2008.10.043 (2008).

- 10 Dubois, J., de Berker, A. O. & Tsao, D. Y. Single-unit recordings in the macaque face patch system reveal limitations of fMRI MVPA. *J Neurosci* **35**, 2791-2802, doi:10.1523/JNEUROSCI.4037-14.2015 (2015).
- 11 Kriegeskorte, N. Relating population-code representations between man, monkey, and computational models. *Frontiers in Neuroscience* **3**, 35 (2009).
- 12 Leutgeb, J. K., Leutgeb, S., Moser, M. B. & Moser, E. I. Pattern separation in the dentate gyrus and CA3 of the hippocampus. *Science* **315**, 961-966 (2007).
- 13 El-Kalliny, M. M. *et al.* Changing temporal context in human temporal lobe promotes memory of distinct episodes. *Nature Communications* **10**, 203, doi:10.1038/s41467-018-08189-4 (2019).
- 14 Kyle, C. T., Stokes, J. D., Lieberman, J. S., Hassan, A. S. & Ekstrom, A. D. Successful retrieval of competing spatial environments in humans involves hippocampal pattern separation mechanisms. *Elife* **4**, e10499, doi:10.7554/eLife.10499 (2015).
- 15 Stevenson, R. F., Reagh, Z. M., Chun, A. P., Murray, E. A. & Yassa, M. A. Pattern Separation and Source Memory Engage Distinct Hippocampal and Neocortical Regions during Retrieval. *J Neurosci* **40**, 843-851, doi:10.1523/JNEUROSCI.0564-19.2019 (2020).
- 16 Chanales, A. J. H., Oza, A., Favila, S. E. & Kuhl, B. A. Overlap among Spatial Memories Triggers Repulsion of Hippocampal Representations. *Curr Biol* **27**, 2307-2317 e2305, doi:10.1016/j.cub.2017.06.057 (2017).
- 17 Wanjia, G., Favila, S. E., Kim, G., Molitor, R. J. & Kuhl, B. A. Abrupt hippocampal remapping signals resolution of memory interference. *Nature Communications* **12**, 4816, doi:10.1038/s41467-021-25126-0 (2021).
- 18 Attwell, D. *et al.* Glial and neuronal control of brain blood flow. *Nature* **468**, 232-243 (2010).
- 19 Canals, S., Beyerlein, M., Murayama, Y. & Logothetis, N. K. Electric stimulation fMRI of the perforant pathway to the rat hippocampus. *Magnetic Resonance Imaging* **26**, 978-986 (2008).
- 20 Logothetis, N. K., Pauls, J., Augath, M., Trinath, T. & Oeltermann, A. Neurophysiological investigation of the basis of the fMRI signal. *Nature* **412**, 150-157 (2001).
- 21 Ekstrom, A. D. Regional variation in neurovascular coupling and why we still lack a Rosetta Stone. *Philos Trans R Soc Lond B Biol Sci* (2020).
- 22 Colgin, L. L., Moser, E. I. & Moser, M. B. Understanding memory through hippocampal remapping. *Trends Neurosci* **31**, 469-477, doi:10.1016/j.tins.2008.06.008 (2008).
- 23 Sakon, J. J. & Suzuki, W. A. A neural signature of pattern separation in the monkey hippocampus. *Proc Natl Acad Sci U S A* **116**, 9634-9643, doi:10.1073/pnas.1900804116 (2019).
- 24 Marr, D., Willshaw, D. & McNaughton, B. in *From the Retina to the Neocortex* 59-128 (Springer, 1991).
- 25 Hunsaker, M. R. & Kesner, R. P. The operation of pattern separation and pattern completion processes associated with different attributes or domains of memory. *Neurosci Biobehav Rev* **37**, 36-58, doi:10.1016/j.neubiorev.2012.09.014 (2013).
- 26 Yassa, M. A. & Stark, C. E. Pattern separation in the hippocampus. *Trends Neurosci* **34**, 515-525, doi:10.1016/j.tins.2011.06.006 (2011).
- 27 Nolan, C. R., Vromen, J. M., Cheung, A. & Baumann, O. Evidence against the detectability of a hippocampal place code using functional magnetic resonance imaging. *eNeuro* **5** (2018).
- 28 Ekstrom, A., Suthana, N., Millett, D., Fried, I. & Bookheimer, S. Correlation Between BOLD fMRI and Theta-Band Local Field Potentials in the Human Hippocampal Area. *Journal of Neurophysiology* **101**, 2668-2678, doi:10.1152/jn.91252.2008 (2009).

REVIEWER COMMENTS

Reviewer #1 (Remarks to the Author):

The authors have significantly added to the manuscript to address the concern I raised about their ability to map from fMRI to “neural codes”. Note, it was more the “codes” aspect than the “neural” aspect that was driving my concern. Regardless, it is a more complete and balanced presentation of the study and I have no further concerns.